# Alpha protocadherins and Pyk2 kinase regulate cortical neuron migration and cytoskeletal dynamics via Rac1 GTPase and WAVE complex in mice

Li Fan[1,2,3†‡], Yichao Lu[1,2,3†], Xiulian Shen[1,2,3], Hong Shao[1,2,3], Lun Suo[1,4], Qiang Wu[1,2,3*]

[1]Key Laboratory of Systems Biomedicine (Ministry of Education), Center for Comparative Biomedicine, Institute of Systems Biomedicine, Shanghai Center for Systems Biomedicine, Shanghai Jiao Tong University, Shanghai, China; [2]State Key Laboratory of Oncogenes and Related Genes, Shanghai Cancer Institute, Renji Hospital affiliated to Shanghai Jiao Tong University Medical School, Shanghai, China; [3]School of Life Sciences and Biotechnology, Shanghai Jiao Tong University, Shanghai, China; [4]Department of Assisted Reproduction, Shanghai Jiao Tong University Medical School, Shanghai, China

**Abstract** Diverse clustered protocadherins are thought to function in neurite morphogenesis and neuronal connectivity in the brain. Here, we report that the protocadherin alpha (*Pcdha*) gene cluster regulates neuronal migration during cortical development and cytoskeletal dynamics in primary cortical culture through the WAVE (Wiskott-Aldrich syndrome family verprolin homologous protein, also known as Wasf) complex. In addition, overexpression of proline-rich tyrosine kinase 2 (Pyk2, also known as Ptk2b, Cakβ, Raftk, Fak2, and Cadtk), a non-receptor cell-adhesion kinase and scaffold protein downstream of Pcdhα, impairs cortical neuron migration via inactivation of the small GTPase Rac1. Thus, we define a molecular Pcdhα/WAVE/Pyk2/Rac1 axis from protocadherin cell-surface receptors to actin cytoskeletal dynamics in cortical neuron migration and dendrite morphogenesis in mouse brain.
DOI: https://doi.org/10.7554/eLife.35242.001

*For correspondence: qiangwu@sjtu.edu.cn

[†]These authors contributed equally to this work

**Present address:** [‡]Developmental Biology Program, Sloan Kettering Institute, New York, United States

**Competing interests:** The authors declare that no competing interests exist.

## Introduction

The human brain contains approximately 86 billion neurons, and each neuron engages in several thousand specific synaptic connections, resulting in complex neural networks with over $10^{15}$ specific connections. These complex neural circuits are required for normal brain function, and inappropriate assemblies of neural circuits underlie neurodevelopmental and neuropsychiatric disorders (*Hyman, 2008*). A remarkable feature of neurodevelopment is the long-distance neuronal migration from the site of origin to the final destination (*Angevine and Sidman, 1961*; *Ayala et al., 2007*). For example, cortical immature neurons generated from the proliferative ventricular and subventricular zones (VZ/SVZ) migrate radially through specific phases to appropriate laminar positions in an 'inside-out' manner and then differentiate into distinct subtypes of cortical neurons (*Angevine and Sidman, 1961*; *LoTurco and Bai, 2006*; *Rakic, 1974*). The cortical migration phases include somal translocation, multipolar migration, and glial-guided locomotion (*Ayala et al., 2007*; *Cooper, 2014*; *Noctor et al., 2004*). Newly born bipolar neurons in SVZ assume multipolar or stellate morphology and migrate randomly in the intermediate zone (IZ), moving tangentially, up, or down (*Ayala et al., 2007*; *Cooper, 2014*; *Jossin and Cooper, 2011*; *Nadarajah et al., 2003*; *Noctor et al., 2004*;

**eLife digest** There are hundreds of billions of neurons in a human brain, and each one can form several thousand connections with other neurons. This complex network determines our thoughts, memories, personality, and behavior, but how does it form? During brain development, specific areas give rise to new neurons, which then migrate long distances to other parts of the brain. Upon arrival, they generate several structures, called dendrites, which connect with other neurons.

To distribute themselves correctly, the migrating immature neurons must be able to travel long distances and steer clear of one another. The dendrites from a single mature neuron must also avoid each other, a phenomenon known as self-avoidance. Certain membrane-spanning proteins, called clustered protocadherins, may help neurons achieve this. The portion of the protocadherins that sits on the cell surface is highly variable, and acts as a zipcode that helps cells to recognize one another. However, the section of the protein inside the cell varies little and is shared by all members of a protocadherin family. When the clustered protocadherin is 'switched on', this internal segment can trigger a cascade of reactions that create changes in the cell. Yet, little was known about the nature of this signaling cascade.

Using gene editing in mice, Fan, Lu et al. focus on the signaling cascade of the clustered protocadherin alpha family. The experiments show that the internal portion of these proteins interacts with a protein complex called WAVE. It also inhibits an enzyme known as Pyk2, which increases the activity of another enzyme called Rac1 GTPase, that then further activates WAVE. This results in the WAVE complex also interacting with the internal skeleton inside the neurons and dendrites, which regulates the ability of these cells to migrate and of the dendrites to avoid each other.

Many brain conditions, such as autism spectrum disorders or depression, result from abnormal neuronal migration and connectivity. Mutations in the genes of clustered protocadherins increase the risk of these disorders. By showing how these proteins help to regulate the migration and connectivity of neurons, Fan, Lu et al. add to our understanding of brain development in health and disease.

DOI: https://doi.org/10.7554/eLife.35242.002

*Tabata and Nakajima, 2003*). They then transit into bipolar again near the border of IZ/CP (cortical plate) and resume final radial migration to settle in appropriate cortical layers (*Ayala et al., 2007*; *Cooper, 2014*; *Jossin and Cooper, 2011*; *Nadarajah et al., 2003*; *Noctor et al., 2004*; *Tabata and Nakajima, 2003*). Abnormal neuronal migration results in various neurodevelopmental and psychiatric diseases (*Ayala et al., 2007*; *LoTurco and Bai, 2006*; *Valiente and Marín, 2010*); however, the underlying molecular mechanisms for the abnormal neuronal migration is largely unknown.

Human genetics studies have implicated mutations of the clustered protocadherin (*Pcdh*) cell adhesion genes in the 5q31 region for various developmental and psychiatric disorders (*Anitha et al., 2013*; *Iossifov et al., 2012*; *Pedrosa et al., 2008*; *Shimojima et al., 2011*). Similar to *Dscam1* in *Drosophila* (*Zipursky and Sanes, 2010*), diverse clustered *Pcdh* genes play an important role in establishing neuronal identity and connectivity in the vertebrate brain (*Garrett et al., 2012*; *Lefebvre et al., 2012*; *Molumby et al., 2016*; *Nicoludis et al., 2016*; *Rubinstein et al., 2015*; *Schreiner and Weiner, 2010*; *Suo et al., 2012*; *Thu et al., 2014*; *Wu and Maniatis, 1999*). In mice, 58 clustered *Pcdh* genes are organized into three closely linked *Pcdh* $\alpha$, $\beta$, and $\gamma$ clusters (*Pcdha*, *Pcdhb*, and *Pcdhg*) (*Wu et al., 2001*). The *Pcdh* $\alpha$ and $\gamma$ clusters are each consisted of variable and constant regions, similar to that of the *Ig*, *Tcr*, and *Ugt1* gene clusters (*Wu, 2005*; *Wu and Maniatis, 1999*; *Wu et al., 2001*; *Zhang et al., 2004*). In particular, the variable region of the mouse *Pcdh*$\alpha$ cluster contains 12 highly similar 'alternate exons', $\alpha 1$-$\alpha 12$, whose promoters are stochastically activated by distal enhancers, and two divergent c-type 'ubiquitous exons', $\alpha c1$ and $\alpha c2$, whose promoters are constitutively activated by distal enhancers (*Figure 1A*) (*Guo et al., 2012*). Each variable exon is separately spliced to the common set of downstream constant exons, generating diverse mRNAs and proteins. CCCTC-binding factor (CTCF)/Cohesin-mediated topological chromatin-looping domains are crucial for proper expression of Pcdh$\alpha$ proteins (*Guo et al., 2015*; *Huang and Wu, 2016*). Each variable exon encodes an extracellular domain (ectodomain EC1-6), a transmembrane

segment, and a juxtamembrane variable cytoplasmic domain (VCD) (*Shonubi et al., 2015*; *Wu and Maniatis, 1999*), whereas the three constant exons encode a common membrane-distal constant domain (CD) of all Pcdhα proteins (*Figure 1A*). This suggests that diverse extracellular cues converge on a single intracellular signaling pathway. However, the functional significance of this intriguing arrangement remains obscure.

A large family of cell-surface receptors, including Pcdhα6 (*Pcdha6*), recruit WAVE complex to the plasma membrane (*Chen et al., 2014*; *Nakao et al., 2008*; *Tai et al., 2010*). The WAVE complex is a conserved two-partite pentameric complex consisting of a pseudosymmetric dimer of Sra1/Cyfip1 and Nap1/Hem2, and a heteromeric trimer of HSPC300/Brick, Abi1/2/3, and WAVE1/2/3/SCAR (*Chen et al., 2010*). First, Abi2 interacts with Abelson tyrosine kinase (Abl kinase) and has been implicated in cortical radial migration (*Xie et al., 2013*). Second, WAVEs/SCARs are members of the Wiskott-Aldrich syndrome protein (WASP) and WASP verprolin homologous protein family, defined by a conserved VCA domain (verprolin homologous, cofilin homologous or central hydrophobic, and acidic regions) (*Chen et al., 2010*). Third, VCA is inhibited by intermolecular interaction with Sra1 and intramolecular interaction within WAVE (*Chen et al., 2010*; *Padrick et al., 2011*; *Rohatgi et al., 1999*). Fourth, Rac1 binds to WAVE complex and induces a conformational change to release VCA from its inhibitory state and to activate actin filament nucleation and branching through the Arp2/3 complex (*Chen et al., 2010*; *Lebensohn and Kirschner, 2009*; *Padrick et al., 2011*; *Rohatgi et al., 1999*; *Ti et al., 2011*). Finally, Pyk2, a calcium-dependent cell-adhesion kinase and scaffold protein highly expressed in the brain and inhibited by Pcdhα, also regulates neurodevelopment (*Chen et al., 2009*; *Hsin et al., 2010*; *Lev et al., 1995*; *Suo et al., 2012*). However, whether and how WAVE complex and Pyk2 kinase function in cortical neuron migration are not clear.

Here, we report that Pcdhα proteins play a critical role in neuronal migration and cytoskeletal dynamics. Specifically, we define an actin cytoskeleton remodeling pathway by which Pcdhα regulates lamellipodial and filopodial dynamics and neuronal migration as well as dendrite morphogenesis through interaction with WAVE complex via the WIRS (WAVE interacting receptor sequence) motif of Pcdhα constant domain (CD). In addition, Pyk2 regulates cortical neuron migration by inactivating the small GTPase Rac1. Given that actin cytoskeletal dynamics are central for neurite morphogenesis and neuronal migration, our findings have interesting implications for mechanisms of *Pcdhα* functions in dendrite self-avoidance and neuronal self/nonself recognition in normal brain development as well as aberrant neuron migration and dendrite morphogenesis underlying complex neurodevelopmental diseases.

## Results

### Defective cortical neuron migration with Pcdhα knockdown

We mapped the embryonic expression pattern of *Pcdhα* by using a GFP knockin mouse line (Pcdhα^GFP) (*Wu et al., 2007*) and found that Pcdhα proteins are expressed throughout the developing forebrain (*Figure 1B*). Immunostaining with an antibody against alpha constant domain (αCD) revealed that Pcdhα proteins are expressed in all cortical regions and most prominently in the intermediate zone and marginal zone (IZ and MZ) of the developing neocortex (*Figure 1C*). RT-PCR with isoform-specific primers showed that, starting at E10, every member of the *Pcdhα* cluster is expressed in the developing brain (*Figure 1—figure supplement 1A*). Pcdhα knockdown (αKD) with two independent shRNAs, each targeting a distinct subdomain of the constant region by in utero electroporation (IUE), revealed a significant decrease of migrating neurons in the cortical plate (CP) and a concomitant increase within the lower intermediate zone, suggesting defects in multipolar migration (*Figure 1D* and *Figure 1—figure supplement 1B*). The αKD multipolar neurons in the intermediate zone display stunted processes, as shown by lucida drawings (*Figure 1E*). Live cell imaging of brain organotypic slice culture demonstrated the slower velocity of multipolar migration of αKD neurons compared to controls (*Figure 1F–H* and *Video 1*). In addition, early born αKD neurons also have migration defects, suggesting that Pcdhα is also required for glia-independent somal translocation (*Figure 1I and J*). This suggests that Pcdhα is required for migration of immature cortical neurons.

To rule out the possibility of altered progenitor proliferation, we labeled αKD mouse brain with BrdU and analyzed cell proliferation. Compared with controls, αKD results in no significant difference

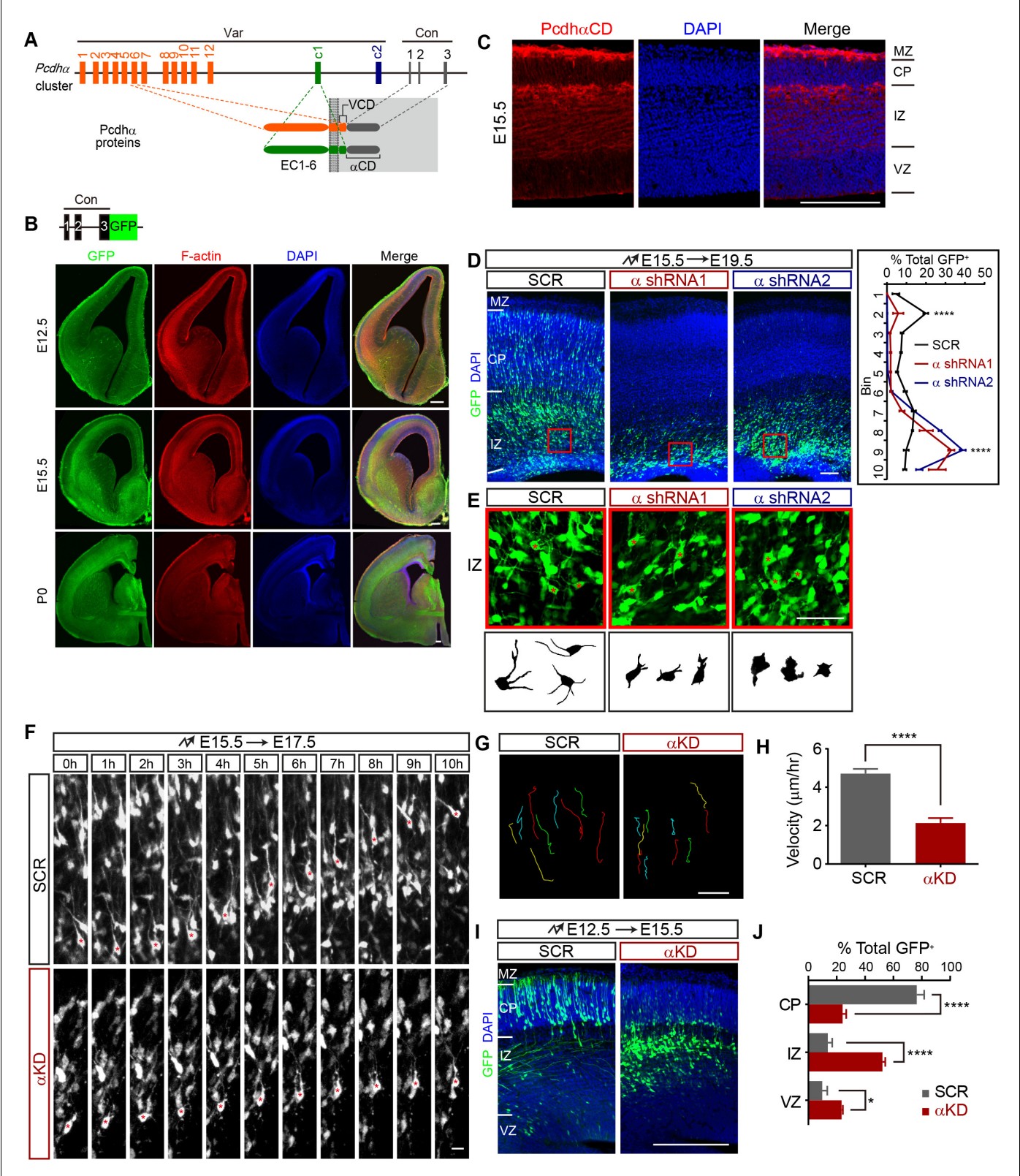

**Figure 1.** Pcdhα is required for cortical neuron migration. (A) Schematics of the mouse Pcdhα organization. Var, variable region; Con, constant region; EC, ectodomain; VCD, variable cytoplasmic domain; αCD, Pcdhα constant domain. (B) GFP and F-actin immunostaining of cortical coronal sections from E12.5, E15.5, and P0 Pcdhα^GFP mouse brain. Nuclei are counterstained with DAPI. Upper left inset shows the Pcdhα^GFP mouse line construction. (C) Immunostaining with an antibody specific for PcdhαCD of cortical coronal sections from E15.5 wild-type mouse brains. Nuclei were counterstained

*Figure 1 continued on next page*

*Figure 1 continued*

with DAPI. (D) Cortical coronal sections of E19.5 mouse brain which were electroporated at E15.5 with control (SCR: scrambled) or αKD (α shRNA1 or α shRNA2) plasmids. Nuclei were counterstained with DAPI. Quantification of GFP⁺ cell distribution across the cortex (divided into ten equal bins) is shown on the right. n = 6 brains for each group. Statistical significance was assessed using one-way ANOVA, followed by a post-hoc Tukey's multiple comparisons test. (E) Representative multipolar neurons and their lucida drawings in the red boxes shown in (D). Asterisks indicate multipolar cells. (F) Embryonic brains were electroporated at E15.5 and organotypic slices were prepared from brains at E17.5. Representative frames from a 10 hr time-lapse imaging experiment are shown. Asterisks indicate one migrating cell. See also *Video 1*. (G and H) Typical migration traces (G) and migration velocity (H) of control and αKD neurons in a time-lapse experiment shown in (F). n = 15 cells for each group. Student's *t* test. (I) Cortical coronal sections of E15.5 embryos electroporated at E12.5 with control or αKD plasmids. Nuclei were counterstained with DAPI. (J) Quantification of E15.5 control and αKD GFP⁺ cells in CP, IZ, and VZ. n = 5 brains for SCR, n = 4 brains for αKD. Student's *t* test. Data as mean ± SEM. ****p<0.0001. *p<0.05. See *Figure 1—source data 1*. Scale bar, 20 μm for (F) and (G); 50 μm for (E); 100 μm for all other panels. MZ, marginal zone; CP, cortical plate; IZ, intermediate zone; SVZ, subventricular zone; VZ, ventricular zone.

DOI: https://doi.org/10.7554/eLife.35242.003

The following source data and figure supplements are available for figure 1:

**Source data 1.** Quantification source data for *Figure 1*.
DOI: https://doi.org/10.7554/eLife.35242.005
**Figure supplement 1.** Additional control data for Pcdhα function in cortical neuron migration.
DOI: https://doi.org/10.7554/eLife.35242.004
**Figure supplement 1—source data 1.** Quantification source data for *Figure 1—figure supplement 1*.
DOI: https://doi.org/10.7554/eLife.35242.006

of percentage of BrdU⁺ cells (*Figure 1—figure supplement 1C and D*). In addition, αKD does not alter the percentage of Tbr2⁺intermediate progenitor cells (IPCs) (*Figure 1—figure supplement 1E and F*), nor the morphology of brain lipid binding protein (BLBP)-labeled radial glia cells (*Figure 1—figure supplement 1G*). Moreover, the defect is not due to increased apoptosis (*Figure 1—figure supplement 1H*). Finally, there is no cortical migration defect (*Figure 1—figure supplement 1I*) in mice with deletion of the entire *Pcdhα* cluster (αKO) (*Wu et al., 2007*). The phenotypic discrepancy may be due to known genetic compensation mechanisms induced by deletion but not knockdown (*Rossi et al., 2015*).

## Rescuing cortical neuron migration by single Pcdhα isoforms and constant domain

To rescue the migration defect, we constructed shRNA-resistant forms of α6 (α6*), which represents members of the alternate α1-α12, and of the two divergent c-types (αc1* and αc2*) (*Figure 2—figure supplement 1A*). Indeed the single α6* isoform rescues the αKD migration defect. The Pcdh αc1* also rescues the migration defect; however, αc2* does not (*Figure 2A* and *Figure 2—figure supplement 1B*). This suggests that αc2 has distinct functions other than cortical neuron migration, consistent with very recent findings that αc2 endows serotonergic neurons with a single cell-type identity and specifically mediates the axonal tiling and assembly of serotonergic neural circuitries (*Chen et al., 2017*).

**Video 1.** Movement of multipolar neurons of control and αKD electroporated cortices. One frame per 15 min. Playback speed seven frames/s. Scale bar, 50 μm.
DOI: https://doi.org/10.7554/eLife.35242.007

To investigate whether the extracellular domain and transmembrane segment play a role in cortical neuron migration, we replaced them with a myristoylation signal to attach the shRNA-resistant intracellular domain (ICD) to the plasma membrane (Myr-α6ICD*, Myr-αc1ICD*, Myr-αc2ICD*) (*Figure 2—figure supplement 1A*). We found that Myr-α6ICD* and Myr-αc1ICD* rescue the migration defect, and Myr-αc2ICD* does not (*Figure 2—figure supplement 1C and D*). This suggests that the intracellular domain of Pcdhα plays an important role in cortical neuron migration. To investigate why Myr-αc2ICD* cannot rescue the migration defect, we constructed an αc2 VCD-deleted form, which is, by definition, a

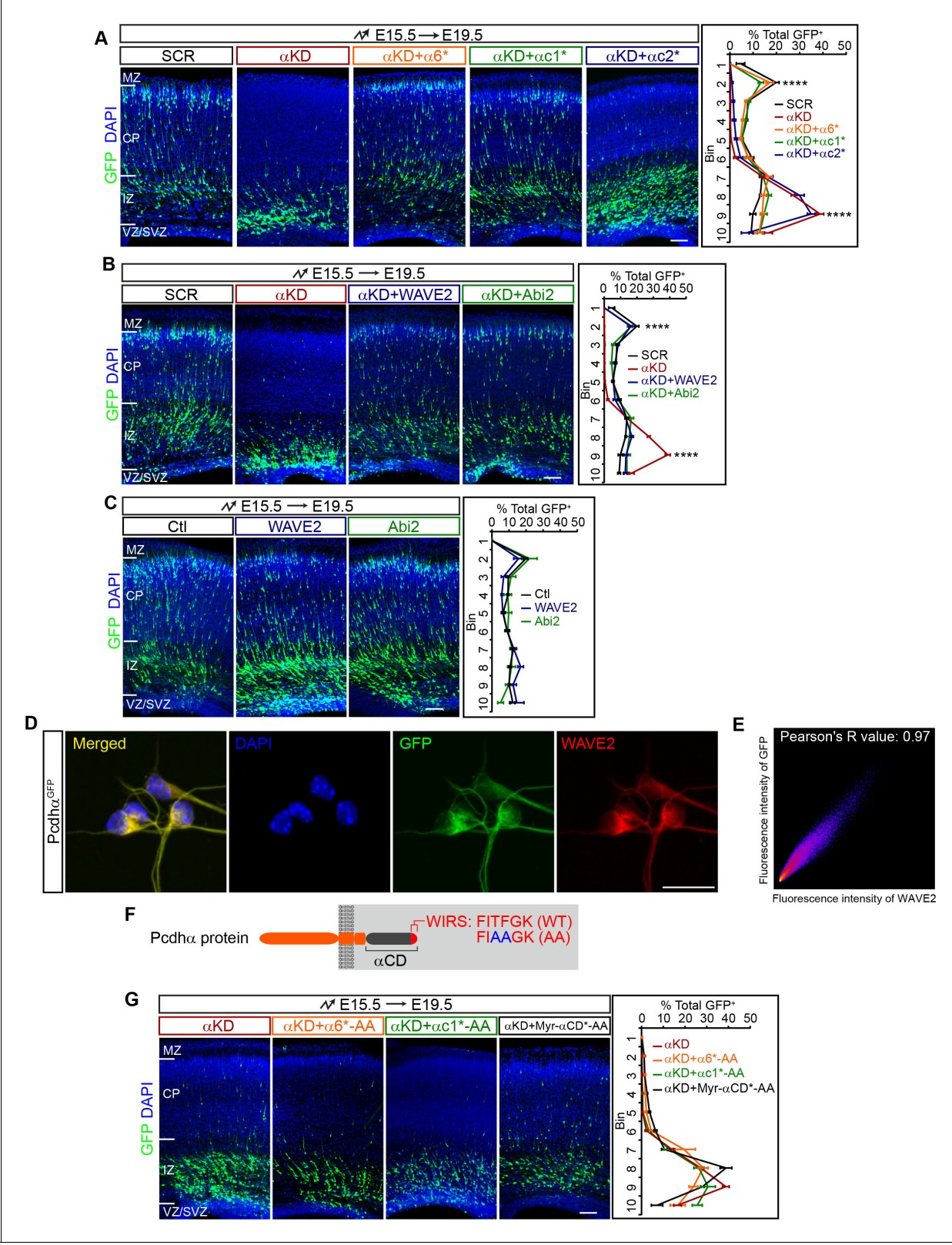

**Figure 2.** Pcdhα regulates cortical neuron migration through the WAVE complex. (A–C) Cortical coronal sections of E19.5 embryonic brains electroporated at E15.5. Nuclei were counterstained with DAPI. Quantification of GFP⁺ cell distribution is shown on the right. n = 6 brains for each group. (D) Immunostaining for GFP and WAVE2 in primary cultured cortical neurons from E17.5 Pcdhα$^{GFP}$ mice. Nuclei were counterstained with DAPI. (E) Two dimensional histogram of Pcdhα and WAVE2 fluorescence intensity in (D). Pearson's R value is analyzed with the ImageJ software. (F)

*Figure 2 continued on next page*

*Figure 2 continued*

Schematics of Pcdhα protein structure with the AA mutation of the WIRS motif. (G) Cortical coronal sections of E19.5 embryonic brains electroporated at E15.5. Nuclei were counterstained with DAPI. Quantification of GFP⁺ cell distribution is shown on the right. n = 6 brains for each group. Data as mean ± SEM. Statistical significance was assessed using one-way ANOVA, followed by a post hoc Tukey's multiple comparisons test. ****p<0.0001. See *Figure 2—source data 1*. Scale bar, 50 µm for (D); 100 µm for all other panels. MZ, marginal zone; CP, cortical plate; IZ, intermediate zone; SVZ, subventricular zone; VZ, ventricular zone.

DOI: https://doi.org/10.7554/eLife.35242.008
The following source data and figure supplements are available for figure 2:

**Source data 1.** Quantification source data for *Figure 2*.
DOI: https://doi.org/10.7554/eLife.35242.010
**Figure supplement 1.** Additional data for Pcdhα function in cortical neuron migration.
DOI: https://doi.org/10.7554/eLife.35242.009
**Figure supplement 1—source data 1.** Quantification source data for *Figure 2—figure supplement 1*.
DOI: https://doi.org/10.7554/eLife.35242.011

myristoylated α constant domain (Myr-αCD*) (*Figure 2—figure supplement 1A*). Intriguingly, we found that Myr-αCD* rescues the migration defect (*Figure 2—figure supplement 1C and D*). This demonstrated that αc2 variable cytoplasmic domain has an inhibitory function. Consistently, sequence analysis revealed that αc2 variable cytoplasmic domain is the longest and most divergent among those of αc1 as well as of α1-α12 (*Figure 2—figure supplement 1E*). Together, these data suggest that members of the Pcdhα family except αc2 regulate cortical neuron migration through their common constant domain.

## Rescuing cortical neuron migration by the WAVE complex

Recent studies linked Pcdhα6 to the WAVE complex through the WIRS (WAVE interacting receptor sequence) motif within the Pcdhα constant domain (*Chen et al., 2014*). We thus investigated whether Pcdhα regulates cortical neuron migration through WAVE. Remarkably, we found that over-expression of either *WAVE2* (*Wasf2*) or *Abi2 in vivo* rescues the cortical neuron migration defect of αKD neurons (*Figure 2B*) although they themselves have no apparent influence on cortical neuron migration (*Figure 2C*). Consistently, endogenous Pcdhα and WAVE2 co-localize in primary cultured cortical neurons (*Figure 2D and E*). In addition, mutating the WIRS motif (from FITFGK to FIAAGK) of α6*, αc1*, and Myr-αCD* (α6*-AA, αc1*-AA, and Myr-αCD*-AA) abolishes the rescue effect (*Figure 2F and G*, in comparison to *Figure 2A* and *Figure 2—figure supplement 1D*). As controls, these WIRS-mutated isoforms as well as wild types appears to reach the plasma membrane (*Figure 2—figure supplement 1F*). Thus, Pcdhα regulates cortical neuron migration through the WAVE complex.

## A role of Pyk2 in cortical neuron migration

Pcdhα physically interacts with and negatively regulates the Pyk2 kinase (*Chen et al., 2009*). In addition, we previously found that Pcdhα regulates dendritic and spine morphogenesis through inhibiting Pyk2 activity (*Suo et al., 2012*). To this end, we investigated whether knockdown of Pyk2 could rescue cortical neuron migration defects of αKD. Although *Pyk2* (*Ptk2b*) knockdown (Pyk2KD) per se or CRISPR knockout of *Pyk2* (Pyk2KO) does not affect cortical neuron migration (*Figure 3A* and *Figure 3—figure supplement 1A*), we found that Pyk2KD rescues the defect of cortical neuron migration in αKD (*Figure 3A* and *Figure 3—figure supplement 1B*). This suggests that Pcdhα regulates cortical neuron migration, at least in part, through the inhibition of Pyk2.

We next asked whether overexpression of Pyk2 (Pyk2OE) could recapitulate αKD cortical neuron migration defects. We found that the majority of Pyk2OE cells are stalled in the middle intermediate zone (mIZ) (*Figure 3B*), a stage little later than the stalling of αKD cells (*Figure 3A*). In addition, these mIZ cells have aberrant multipolar morphology with supernumerary primary processes in comparison to single leading processes of control cells (*Figure 3C–E*). For the very few Pyk2OE cells in the lower cortical plate (CP), they harbor elaborated leading processes (*Figure 3F and G*); by contrast, control cells displayed typical bipolar morphology with a single or bifurcated thick leading process (*Figure 3F and G*). Pyk2OE leads to the inhibition of Rac1 activity (*Suo et al., 2012*). As Rac1 is thought to provide the spatial information for actin polymerization (*Tahirovic et al., 2010*), loss of

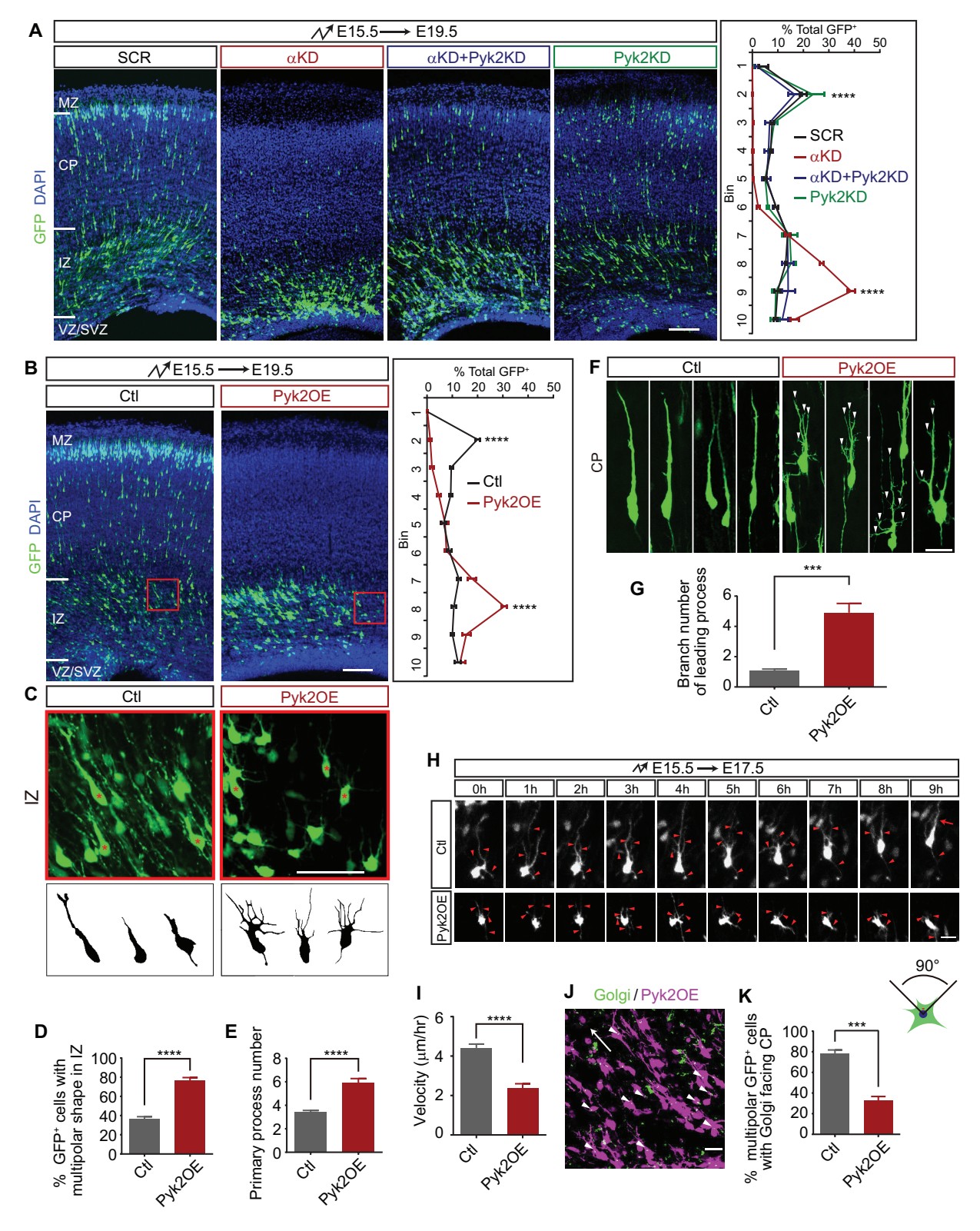

**Figure 3.** Pyk2 regulates cortical neuron migration. (**A**) Cortical coronal sections of E19.5 embryonic brains electroporated at E15.5. Nuclei were counterstained with DAPI. Quantification of GFP+ cell distribution is shown on the right. n = 6 brains for each group. Statistical significance was assessed using one-way ANOVA, followed by a post hoc Tukey's multiple comparisons test. (**B**) Cortical coronal sections of E19.5 embryonic brains electroporated at E15.5 with control or Pyk2-overexpressing (Pyk2OE) plasmids. Nuclei were counterstained with DAPI. Quantification of GFP+ cell

*Figure 3 continued on next page*

*Figure 3 continued*

distribution is shown on the right. n = 6 brains for each group. (C) High magnification of cortical neurons in the red boxes shown in (B). Lucida drawings are shown in the lower panels. (D) Percentage of GFP[+] cells with multipolar morphology in IZ of control and Pyk2OE groups shown in (B). n = 6 brains for each group. (E) Primary process number per cell in IZ of control and Pyk2OE groups shown in (B). n = 10 cells for each group. (F) Typical cortical plate neuron morphology of control and Pyk2OE groups shown in (B). Arrowheads, aberrant branching leading processes. (G) Branch number of leading processes per cortical plate neuron of control and Pyk2OE groups shown in (F). n = 11 cells for each group. (H) Embryonic brains were electroporated *in utero* with control or Pyk2OE plasmids at E15.5. The organotypic slices are cut at E17.5. Representative frames from a 9 hr time-lapse are shown. Arrowheads, neurites; Arrow, leading process. See also *Video 2*. (I) Quantification of the migration velocity of control and Pyk2OE neurons. n = 19 cells for each group. (J) Golgi staining (green, arrowheads) of Pyk2OE neurons (magenta) in IZ at E19.5. Arrow indicates the orientation to CP. (K) Percentage of cells with Golgi facing the CP of control and Pyk2OE groups. n = 3 sections for each group. Data as mean ± SEM. Student's *t* test for (B), (D), (E), (G), (I), (K); ***p<0.001; ****p<0.0001. See *Figure 3—source data 1*. Scale bar, 100 μm for (A, B); 50 μm for (C); 20 μm for all other panels. MZ, marginal zone; CP, cortical plate; IZ, intermediate zone; SVZ, subventricular zone; VZ, ventricular zone.

DOI: https://doi.org/10.7554/eLife.35242.012

The following source data and figure supplements are available for figure 3:

**Source data 1.** Quantification source data for *Figure 3*.

DOI: https://doi.org/10.7554/eLife.35242.014

**Figure supplement 1.** Additional data for Pyk2 function in cortical neuron migration.

DOI: https://doi.org/10.7554/eLife.35242.013

**Figure supplement 1—source data 1.** Quantification source data for *Figure 3—figure supplement 1*.

DOI: https://doi.org/10.7554/eLife.35242.015

Rac1 activity leads to aberrant actin polymerization at many sites with no controlled spatial information, resulting in supernumerary primary processes (*Figure 3C–E*) and more branchy morphology (*Figure 3F and G*). Finally, time-lapse imaging showed that there is a significant difference of velocity of cortical neuron migration between Pyk2OE and control cells (*Figure 3H and I*, and *Video 2*). These data suggest that Pyk2OE partially recapitulates cortical neuron migration defects.

We next examined the orientation of the Golgi apparatus of cells in mIZ, which is essential for transporting vesicles for oriented motility (*Jossin and Cooper, 2011*), by immunostaining with a Golgi marker GM130 (*Figure 3J*). Most Golgi complexes are normally localized in front of the cell nucleus and are oriented toward the cortical plate (*Jossin and Cooper, 2011*). However, the polarity of most Pyk2OE cells is disrupted, showing oblique or inverted orientation of the Golgi apparatus (*Figure 3J and K*). Thus, Pyk2OE blocks multipolar-bipolar transition by disrupting proper localization of the Golgi apparatus. Finally, early-born Pyk2OE neurons are also stalled at the intermediate zone, suggesting that Pyk2 also plays a role in somal translocation (*Figure 3—figure supplement 1C and D*).

To rule out the potential nonspecific effect of the CAG promoter, which is active in both progenitors and postmitotic neurons, we ectopically overexpressed Pyk2 at E15.5 only in postmitotic neurons using the NeuroD promoter (*Jossin and Cooper, 2011*). We found that Pyk2OE under the NeuroD promoter also significantly impairs cortical neuron migration in postmitotic neurons (*Figure 3—figure supplement 1E-G*). Taken together, this suggests that Pcdhα regulates cortical neuron migration, at least in part, through inhibiting Pyk2 kinase activity.

## Regulation of cortical neuron migration by Pyk2 via Rac1

We previously found that Rac1 is epistatic downstream of Pyk2 in dendrite development and spine morphogenesis (*Suo et al., 2012*). To investigate whether Pyk2-Rac1 pathway also functions in cortical neuron migration, we overexpressed a constitutive active form Rac1 (Rac1$^{Q61L}$) in

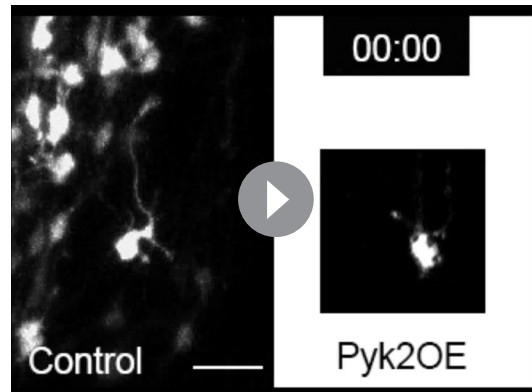

**Video 2.** Movement of bipolar neurons of control and Pyk2OE electroporated cortices. One frame per 15 min. Playback speed seven frames/s. Scale bar, 10 μm
DOI: https://doi.org/10.7554/eLife.35242.016

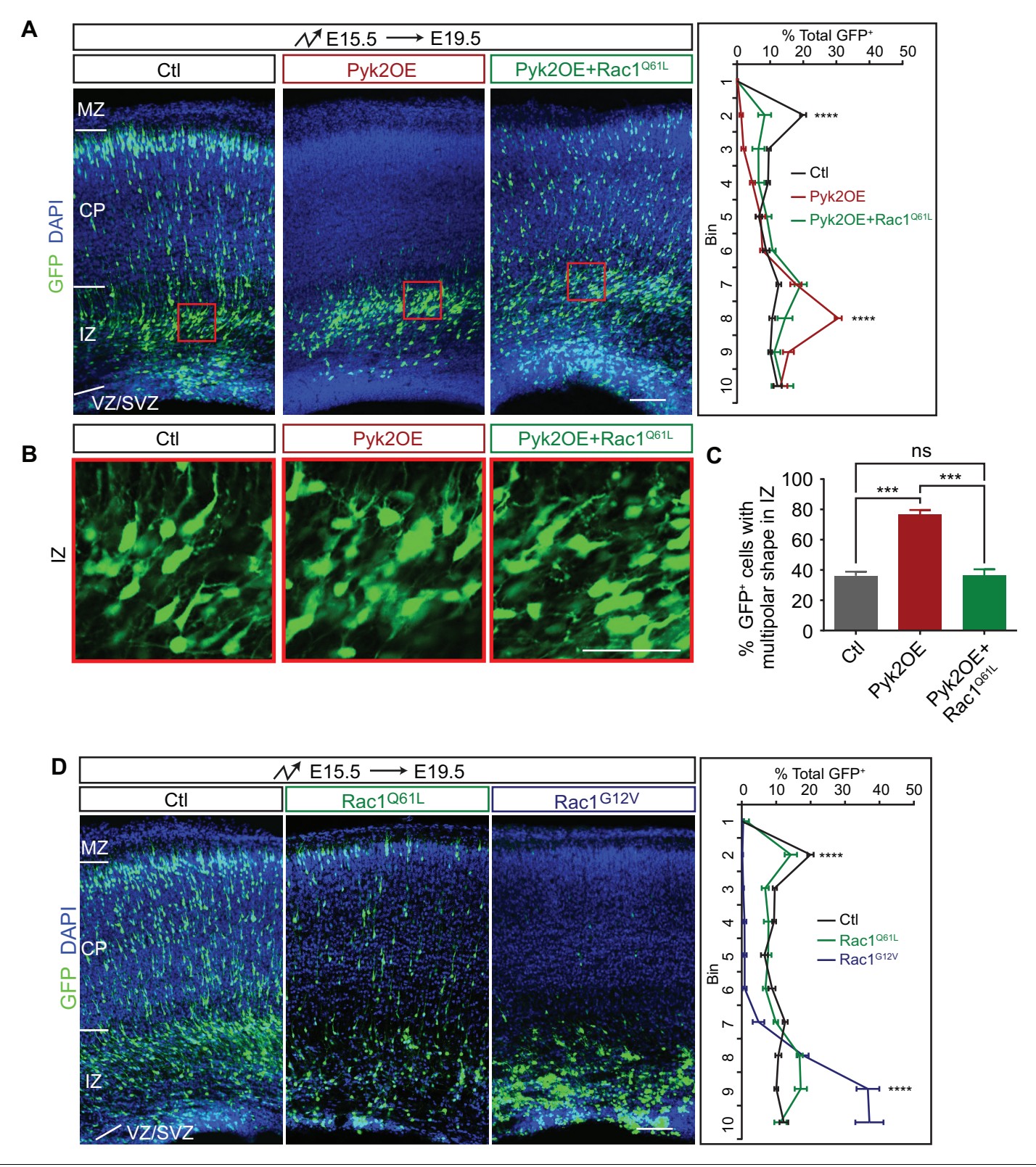

**Figure 4.** Pyk2 regulates cortical neuron migration through Rac1 inhibition. (**A**) Cortical coronal sections of E19.5 embryonic brains electroporated in utero at E15.5. Nuclei were counterstained with DAPI. Quantification of GFP+ cell distribution is shown on the right. n = 6 brains for each group. (**B**) High magnification of cortical neurons in the red boxes shown in (**A**). (**C**) Percentage of multipolar neurons in the IZ region. n = 6 brains for each group. (**D**) Cortical coronal sections of E19.5 embryos electroporated at E15.5. Nuclei were counterstained with DAPI. Quantification of GFP+ cell distribution is

*Figure 4 continued on next page*

*Figure 4 continued*

shown on the right. n = 6 brains for each group. Data as mean ± SEM. Statistical significance was assessed using one-way ANOVA, followed by a post hoc Tukey's multiple comparisons test. ns, not significant; ***p<0.001; ****p<0.0001. See *Figure 4—source data 1*. Scale bar, 50 μm for (**B**); 100 μm for other panels. MZ, marginal zone; CP, cortical plate; IZ, intermediate zone; SVZ, subventricular zone; VZ, ventricular zone.

DOI: https://doi.org/10.7554/eLife.35242.017

The following source data and figure supplements are available for figure 4:

**Source data 1.** Quantification source data for *Figure 4*.

DOI: https://doi.org/10.7554/eLife.35242.019

**Figure supplement 1.** Analyses of Pyk2-domain requirement in cortical neuron migration.

DOI: https://doi.org/10.7554/eLife.35242.018

**Figure supplement 1—source data 1.** Quantification source data for *Figure 4—figure supplement 1*.

DOI: https://doi.org/10.7554/eLife.35242.020

Pyk2OE neurons. We found that Rac1$^{Q61L}$ rescues defects of multipolar migration and morphology of Pyk2OE neurons (*Figure 4A–C*), although Rac1$^{Q61L}$ itself has no apparent effect on cortical neuron migration (*Figure 4D*). However, overexpression of another constitutively active form of Rac1 (Rac1$^{G12V}$) impairs cortical neuron migration (*Figure 4D*) (*Konno et al., 2005*) and cannot be used to rescue, likely because it has a lower affinity for GTP and thus lower constitutive activity than Rac1$^{Q61L}$. Thus, the two constitutively active forms of Rac1 have distinct roles in cortical neuron migration (*Figure 4A and D*). Together, we conclude that Pyk2OE inhibits multipolar-bipolar transition and leads to aberrant branchy morphology in the intermediate zone by inactivating the small GTPase Rac1.

## Dissection of Pyk2 domain in cortical neuron migration

Pyk2 functions as an enzyme through its middle kinase domain and as a molecular scaffold through its N-terminal FERM (four-point-one, ezrin, radixin, moesin) domain (*Figure 4—figure supplement 1A*) (*Chen et al., 2009*; *Lev et al., 1995*; *Suo et al., 2012*). We systematically engineered Pyk2 by mutating a series of key residues of its enzymatic kinase cascade. We found that overexpression of Pyk2$^{Y402F}$, an autophosphorylation mutant that still can be activated by endogenous Pyk2, as well as Pyk2$^{Y579F}$, Pyk2$^{Y580F}$, and Pyk2$^{Y881F}$, still recapitulate the migration defects of αKD (*Figure 4—figure supplement 1A and B*). However, overexpression of Pyk2$^{K457A}$, which has a mutation at the catalytic center and is completely kinase-dead (*Suo et al., 2012*), cannot recapitulate the migration defects of αKD (*Figure 4—figure supplement 1A and B*). This suggests that the catalytic activity of overexpressed Pyk2 is essential for recapitulating the migration defects of αKD.

Remarkably, overexpression of the Pyk2 FERM domain alone recapitulates the blocking activity of Pyk2OE (*Figure 4—figure supplement 1A and C*), whereas deletion of FERM domain abolishes the blocking (*Figure 4—figure supplement 1A and C*). Consistently, the C-terminal FAT domain of Pyk2 is not required for the blocking effect and the kinase domain alone cannot block cortical neuron migration (*Figure 4—figure supplement 1A and C*). This is consistent with that Pyk2 has important kinase-independent functions in contextual fear memory (*Suo et al., 2017*). Together, we conclude that both Pyk2 kinase cascade and FERM scaffold are crucial for blocking cortical neuron migration.

As stated above, constitutive active Rac1$^{Q61L}$ rescues the blocking effect of Pyk2OE (*Figure 4A*). However, we found that Rac1$^{Q61L}$ cannot rescue the blocking activity of FERM domain (*Figure 4—figure supplement 1D*). This suggests that constitutive active form of Rac1 only functions downstream of the kinase cascade but not the FERM scaffold of Pyk2.

## Pcdhα in lamellipodial formation and cytoskeletal dynamics

We next investigated actin dynamics underlying neuronal migration in primary cultured cortical neurons. The early development of primary cultured neurons can be divided into two stages: stage 1, in which the cell body is surrounded by flattened lamellipodia and stage 2, in which the lamellipodia transform into definitive processes with growth cones (*Dotti et al., 1988*). At stage 1, we found that the size of lamellipodia around cell cortex in αKD neurons decreases significantly compared with controls (*Figure 5A and B*). In addition, α6*, αc1*, or Myr-αCD* rescues the αKD lamellipodial

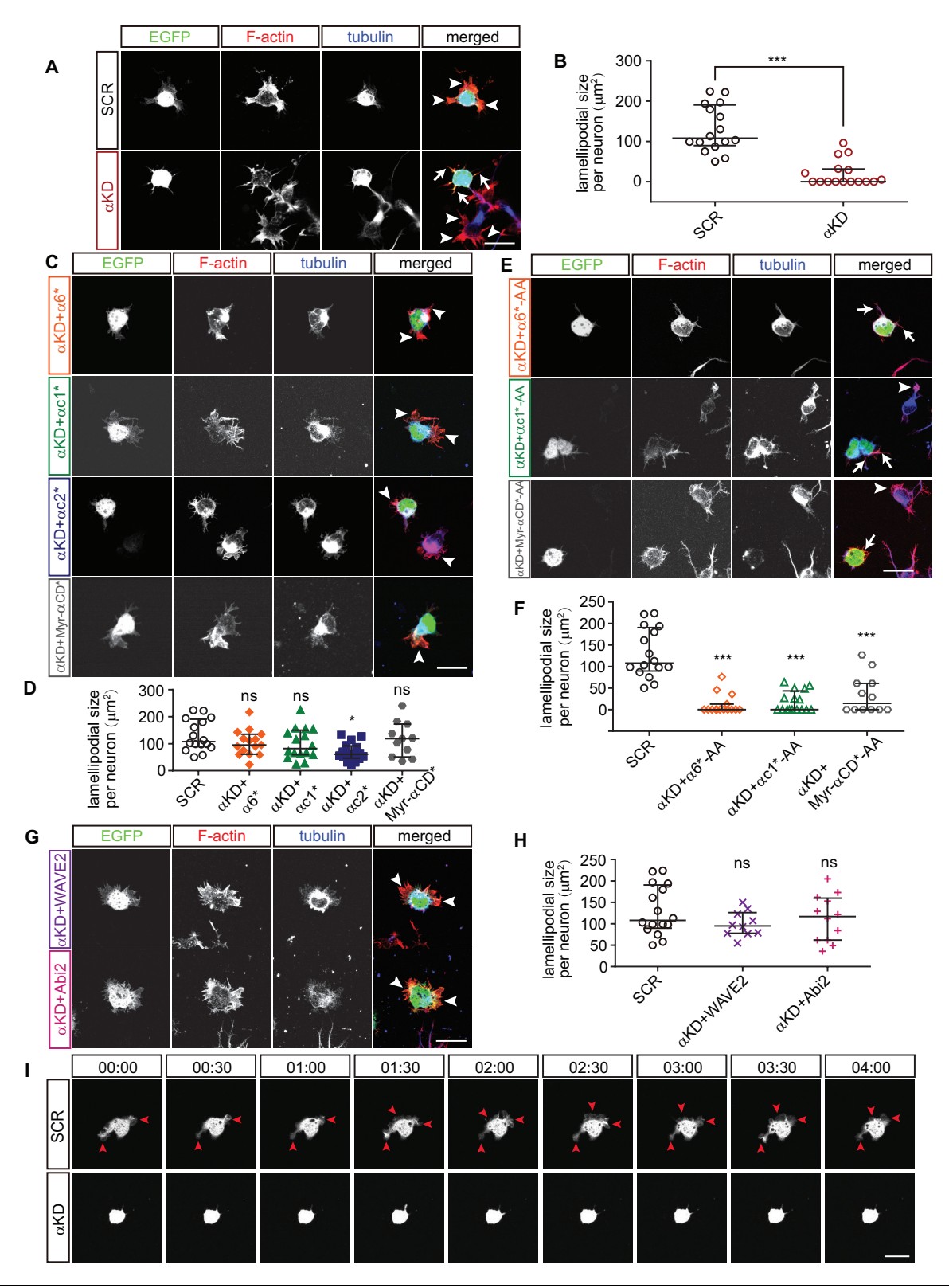

**Figure 5.** Pcdhα regulates lamellipodial dynamics. (A, C, E, G) Primary cultured cortical neurons from E17.5 embryonic cortices, electroporated at E15.5, were in-vitro cultured for 24 hr, immunostained by Tuj1 antibody for tubulin and counterstained with phalloidin for F-actin. Arrowheads, lamellipodia; Arrows, defective lamellipodia. (B, D, F, H) Quantification of lamellipodial size per neuron shown in (A), (C), (E), (G). n = 16 cells for SCR, αKD, αKD+αc1*, αKD+αc1*-AA, and αKD+αc2*; n = 15 cells for αKD+α6*, αKD+α6*-AA; n = 12 cells for αKD + Myr-αCD*-AA, αKD + Abi2; n = 11

*Figure 5 continued on next page*

*Figure 5 continued*

cells for αKD + Myr-αCD*; n = 10 cells for αKD + WAVE2. (**I**) Representative frames from time-lapse imaging of primary cortical neurons cultured in vitro for 24 hr. See also *Video 3*. Arrowheads, lamellipodia. All data are presented as a scatter-dot plot. The median is shown as a line with the interquartile range. Student's *t* test for (**A**). For (**D**), (**F**), (**H**), statistical significance was assessed using one-way ANOVA, followed by a post hoc Tukey's multiple comparisons test. ***p<0.001; ns, not significant. See *Figure 5—source data 1*. Scale bar, 10 μm.

DOI: https://doi.org/10.7554/eLife.35242.021

The following source data and figure supplements are available for figure 5:

**Source data 1.** Quantification source data for *Figure 5*.

DOI: https://doi.org/10.7554/eLife.35242.023

**Figure supplement 1.** Pcdhα is required for lamellipodial formation in stage 2 primary cultured cortical neurons.

DOI: https://doi.org/10.7554/eLife.35242.022

**Figure supplement 1—source data 1.** Quantification source data for *Figure 5—figure supplement 1*.

DOI: https://doi.org/10.7554/eLife.35242.024

defect. By contrast, αc2* does not rescue (*Figure 5C and D*), which is consistent with that αc2* cannot rescue the defects of cortical neuron migration (*Figure 2A*). Moreover, mutating the WIRS motif (from FITFGK to FIAAGK) in either α6*, αc1*, or Myr-αCD* abolishes their rescue effects (*Figure 5E and F*), similar to the situation in cortical neuron migration (*Figure 2G*). Finally, both WAVE2 and Abi2 rescue the lamellipodial defect (*Figure 5G and H*).

At stage 2, αKD results in a significant decrease of percentage of primary neurites with lamellipodia-like protrusions (*Figure 5—figure supplement 1A and B*). Consistent with the situation at stage 1, α6*, αc1*, or Myr-αCD* rescues this αKD lamellipodial defect while αc2* does not (*Figure 5—figure supplement 1C and D*), and mutating the WIRS motif (from FITFGK to FIAAGK) abolishes the rescue effects of either α6*, αc1*, or Myr-αCD* (*Figure 5—figure supplement 1E and F*). In addition, consistent with stage 1, both WAVE2 and Abi2 rescue the lamellipodial defect of stage 2 αKD neurons (*Figure 5—figure supplement 1G and H*).

Finally, αKD lamellipodial dynamics are significantly impaired in comparison with control neurons, whose veil-like lamellipodia are motile and are constantly extending and retracting in both stage 1 and stage 2 neurons (*Figure 5I*, *Figure 5—figure supplement 1I*, *Video 3* and *Video 4*). These data demonstrated that Pcdhα is indispensable for lamellipodial dynamics. Because lamellipodial dynamics are essential for cell migration (*Krause and Gautreau, 2014*), this suggests that cortical neuron migration defects of αKD are a consequence of impairment of lamellipodial formation and cytoskeletal dynamics.

## A comparison between PcdhαKD and Pyk2OE in cytoskeletal dynamics

Consistent with that Pyk2KD rescues cortical neuron migration defects of PcdhαKD (*Figure 3A*), we found that knockdown of Pyk2 in αKD cells

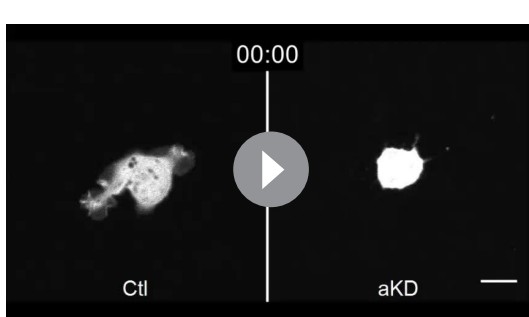

**Video 3.** Dynamics of stage1 control and αKD primary cultured cortical neurons. One frame per 5 min. Playback speed seven frames/s. Scale bar, 20 μm

DOI: https://doi.org/10.7554/eLife.35242.025

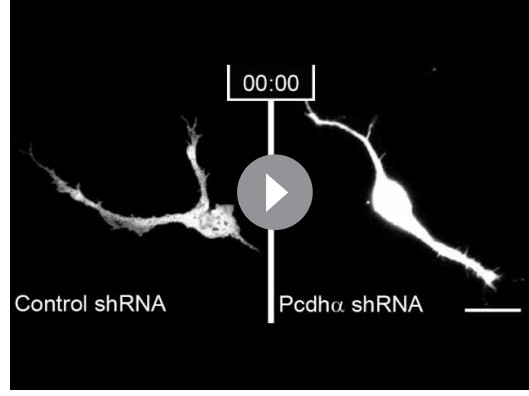

**Video 4.** Dynamics of stage2 control and αKD primary cultured cortical neurons. One frame per 5 min. Playback speed seven frames/s. Scale bar, 40 μm

DOI: https://doi.org/10.7554/eLife.35242.026

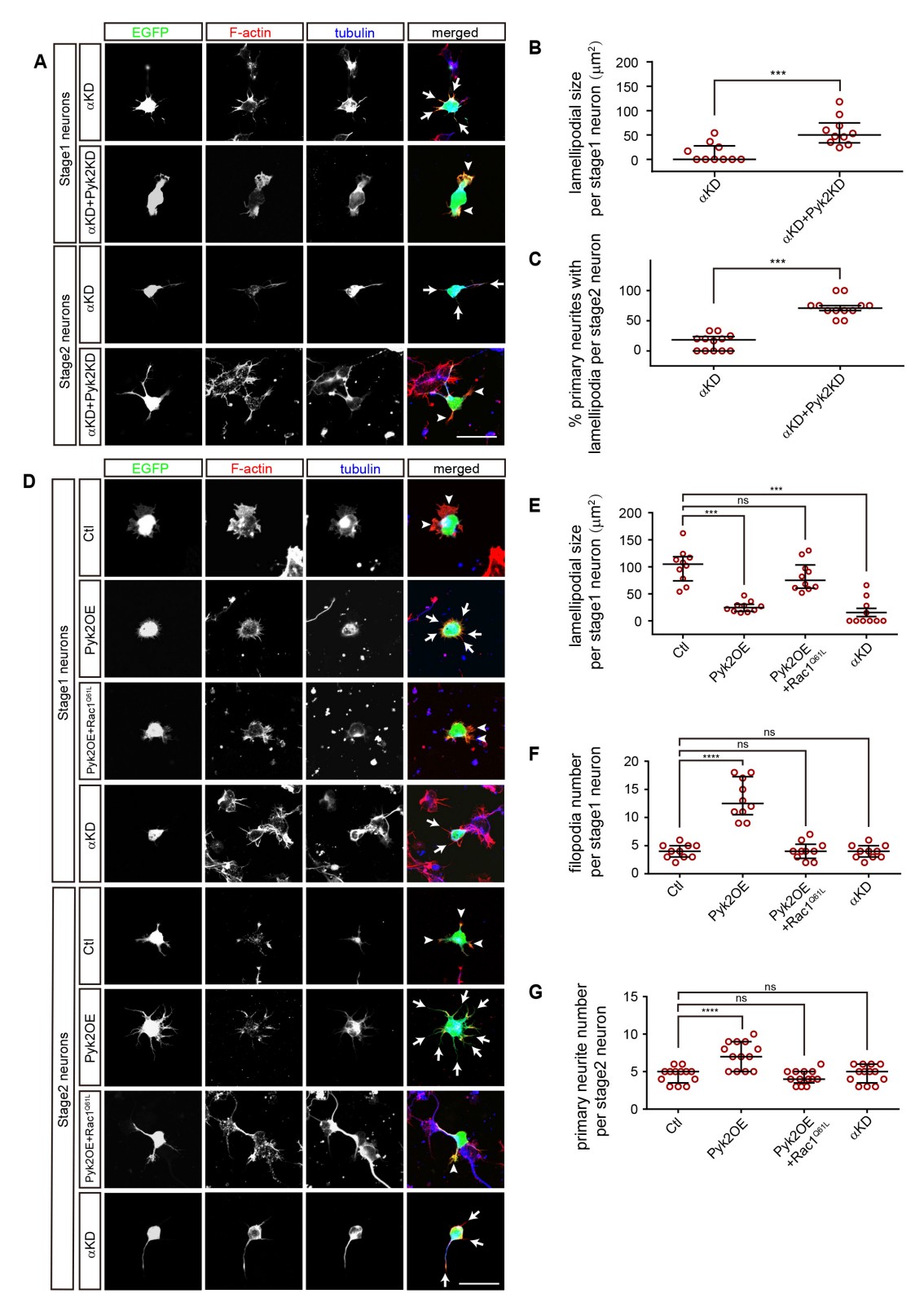

**Figure 6.** A comparison between PcdhαKD and Pyk2OE in cytoskeletal dynamics. (**A**) Primary cultured cortical neurons, derived from E17.5 embryonic cortices which were electroporated at E15.5 with αKD or αKD + Pyk2 KD plasmids, were in-vitro cultured for 24 hr and immunostained by a Tuj1 antibody for tubulin, counterstained with phalloidin for F-actin. Arrowheads, lamellipodia; Arrows, defective lamellipodia. (**B**) Quantification of lamellipodial size per stage 1 neuron shown in **A**). Student's *t* test; n = 10 cells for both groups. (**C**) Percentage of primary neurites with lamellipodia per

*Figure 6 continued on next page*

Figure 6 continued

stage 2 neuron. Student's *t* test; n = 12 cells for both groups. (D) Primary cultured cortical neurons, derived from E17.5 embryonic cortices which were electroporated at E15.5 with indicated plasmids, were in-vitro cultured for 24 hr and immunostained by a Tuj1 antibody for tubulin, counterstained with phalloidin for F-actin. Arrowheads, lamellipodia; Arrows, filopodia. (E) Quantification of lamellipodial size per stage1 neuron shown in (D). Statistical significance was assessed using one-way ANOVA, followed by a post hoc Tukey's multiple comparisons test. n = 10 cells for each group. (F) Quantification of filopodial number per stage 1 neuron shown in (D). Statistical significance was assessed using one-way ANOVA, followed by a post hoc Tukey's multiple comparisons test. n = 10 cells for each group. (G) Quantification of primary neurite number per stage 2 neuron shown in (D). Statistical significance was assessed using one-way ANOVA, followed by a post hoc Tukey's multiple comparisons test. n = 13 cells for each group. All data are presented as a scatter-dot plot. The median is shown as a line with the interquartile range. ****p<0.0001; ***p<0.001; ns, not significant. See *Figure 6—source data 1*. Scale bar, 10 μm.

DOI: https://doi.org/10.7554/eLife.35242.027

The following source data and figure supplement are available for figure 6:

**Source data 1.** Quantification source data for *Figure 6*.

DOI: https://doi.org/10.7554/eLife.35242.029

**Figure supplement 1.** Lifeact-labeled actin cytoskeletal structures in vivo.

DOI: https://doi.org/10.7554/eLife.35242.028

results in a significant increase of lamellipodial sizes of stage1 neurons as well as of the percentage of primary neurites with lamellipodia of stage2 neurons (*Figure 6A–C*). In addition, Pyk2OE results in a significant decrease of lamellipodial sizes, consistent with that of αKD (*Figure 6D and E*).

Filopodia are thin membrane protrusion pushed by underlying actin bundles and filopodial formation is also dependent on Arp2/3 complex (*Mattila and Lappalainen, 2008*), we found that Pyk2OE results in a significant increase of filopodial number per stage 1 neuron as well as of primary neurite number per stage 2 neuron despite no alternation in αKD cells (*Figure 6D–G*). Finally, similar to the rescue of cortical neuron migration defects of PykOE (*Figure 4A*), we found Rac1$^{Q61L}$ rescues both lamellipodial and filopodial defects of Pyk2OE (*Figure 6D–G*). In summary, although both αKD and Pyk2OE impact cytoskeletal dynamics, they have subtle differences on both lamellipodia and filopodia.

To see whether growth cones with lamellipodia and filopodia are affected in vivo, we co-electroporated Lifeact, an actin marker, with either αKD or Pyk2OE plasmids into the developing mouse cortex. In the lower intermediate zone, αKD neurons exhibit abnormal enrichment of Lifeact-labeled actin structures in stunted processes and cell bodies, while the control neurons extend long processes with growth cones (*Figure 6—figure supplement 1A*). In the upper intermediate zone, Pyk2OE neurons exhibit branchy morphology with multiple aberrant processes; however, the control neurons have normal bipolar morphology with single leading processes and growth cones (*Figure 6—figure supplement 1B*).

## Discussion

Recent studies revealed that a zipper-like ribbon structure assembles from combinatorial *cis*- and *trans*-interactions between like-sets of the clustered Pcdhs located in apposed plasma membranes of neighboring cells (*Nicoludis et al., 2016*; *Rubinstein et al., 2015*; *Schreiner and Weiner, 2010*; *Thu et al., 2014*; *Wu, 2005*). These protocadherin interactions could provide enormous diversity and exquisite specificity for neuronal connectivity and neurite self-avoidance required for mammalian brain development. While exquisite specificity is determined by strict homophilic *trans*-interactions of highly diversified EC2/3 (*Goodman et al., 2017*; *Molumby et al., 2016*; *Nicoludis et al., 2016*; *Rubinstein et al., 2015*; *Schreiner and Weiner, 2010*; *Thu et al., 2014*; *Wu, 2005*); enormous diversity is mainly generated by promiscuous *cis*-interactions of highly conserved EC5/6 (*Nicoludis et al., 2016*; *Rubinstein et al., 2015*; *Schreiner and Weiner, 2010*; *Thu et al., 2014*; *Wu, 2005*). One intriguing genomic architecture of the *Pcdhα* cluster is multiple tandem variable exons followed by a single set of three constant exons, encoding a common cytoplasmic constant domain, which is shared by all members of the Pcdhα family (*Figure 1A*) (*Huang and Wu, 2016*; *Wu and Maniatis, 1999*). The extracellular domains of Pcdhα provide enormous diversity and exquisite specificity for cell recognition and adhesion (*Nicoludis et al., 2016*; *Rubinstein et al., 2015*; *Schreiner and*

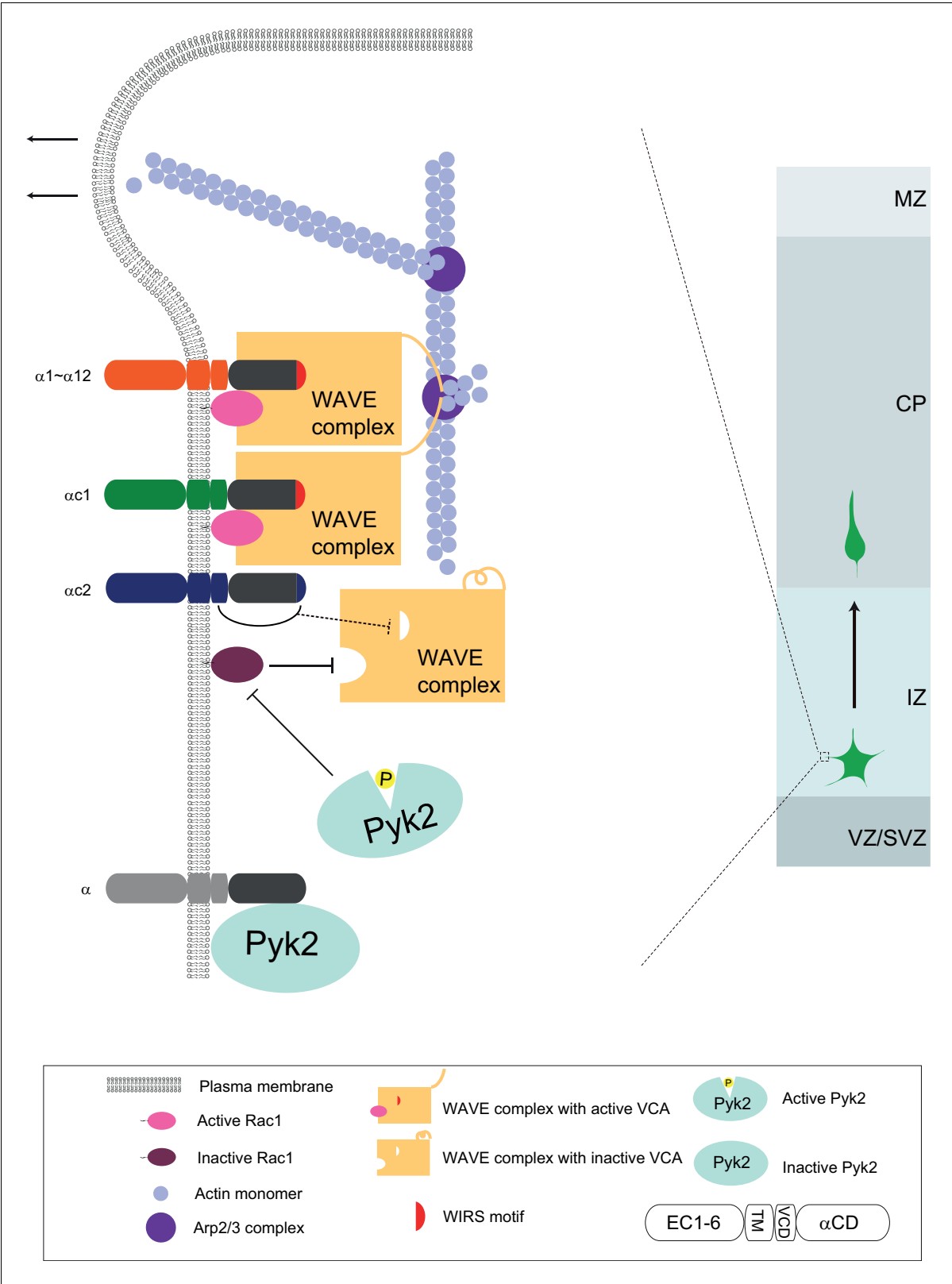

**Figure 7.** A working model of WAVE clustering by protocadherins for actin cytoskeletal dynamics and cortical neuron migration as well as dendrite morphogenesis. On the neuron surface, Pcdhα family proteins recruit WAVE complex to the plasma membrane via the WIRS motif in the Pcdhα constant domain. In addition, Pcdhα proteins also bind to the Pyk2 kinase and inactivate it, thus disinhibits the small GTPase Rac1. The disinhibited Rac1 activates the WAVE complex by inducing a conformation change to release the VCA domains, which are required to activate actin-nucleation by

*Figure 7 continued on next page*

*Figure 7 continued*

the Arp2/3 complex, leading to actin filament branching as well as lamellipodial and filopodial formation. Finally, this protocadherin Pyk2/Rac1/WAVE axis is central for actin cytoskeletal dynamics and cortical neuron migration. The distinct variable cytoplasmic domain of Pcdhαc2 may cause it non-functional for cortical neuron migration. This WAVE clustering model may be a general mechanism for diverse functions of alpha protocadherins in dendrite self-avoidance and neuronal self/nonself recognition as well as dendrite morphogenesis as demonstrated by Sholl analyses (*Figure 7—figure supplement 1*).

DOI: https://doi.org/10.7554/eLife.35242.030

The following source data and figure supplements are available for figure 7:

**Figure supplement 1.** Sholl analysis for protocadherin WAVE-interacting sequence motif in dendrite morphogenesis.

DOI: https://doi.org/10.7554/eLife.35242.031

**Figure supplement 1—source data 1.** Quantification source data for Sholl analyses.

DOI: https://doi.org/10.7554/eLife.35242.032

*Weiner, 2010*; *Thu et al., 2014*; *Wu, 2005*). However, the intracellular Pcdhα signaling pathway is largely unknown.

We propose a Pcdhα-based WAVE clustering model for cortical neuron migration (*Figure 7*). Distinct Pcdhα isoforms on the cell surface recruit WAVE complex to the cell cortex under the plasma membrane. This is strongly supported by (1) the specific interaction between members of the Pcdhα family and the WAVE complex through the WIRS motif in Pcdhα constant domain (*Chen et al., 2014*); (2) the rescue of migration and lamellipodial defects of αKD neurons by WAVE complex subunits WAVE2 and Abi2; and (3) the abolishment of the rescue effect by WIRS mutations. The WIRS motif of members of the Pcdhα family binds to a composite surface formed by Abi2 and Sra1 of the WAVE complex (*Chen et al., 2014*). In addition, the Pcdhα proteins may also recruit WAVE complex through the direct binding of Abi2 C-terminal SH3 domain to the four protocadherin PXXP motifs, which are specific to the constant domain of the Pcdhα but not Pcdhγ family (*Wu and Maniatis, 1999*). Consistently, WAVE2 and Abi2 are required for growth cone activity during cortical neuron migration (*Xie et al., 2013*).

We recently found that N-WASP, a homolog of WAVE2, also regulates cortical neuron migration (*Shen et al., 2018*). In addition, Pcdhα binds to Pyk2 via the intracellular domain and inhibits Pyk2 phosphorylation and activation (*Chen et al., 2009*; *Suo et al., 2012*), resulting in disinhibition of small GTPase Rac1 (*Figure 7*). Moreover, our data suggest that Pyk2 also has kinase-independent scaffolding activity through its FERM (four-point-one, ezrin, radixin, moesin) domain, similar to the FERM domain of FAK, which binds numerous interacting partners and connects cell cortex to diverse downstream intracellular pathways (*Frame et al., 2010*). Rac1, in conjunction with Pcdhα, activate the WAVE complex (*Chen et al., 2010*; *Lebensohn and Kirschner, 2009*; *Rohatgi et al., 1999*). Two activated WAVE complexes, probably induced by protocadherin dimerization, in turn stimulate actin-nucleating activity of Arp2/3 through the two VCAs (*Padrick et al., 2011*; *Ti et al., 2011*). The Arp2/3-mediated actin branching nucleation is central for cytoskeletal dynamics and cell motility (*Krause and Gautreau, 2014*; *Lebensohn and Kirschner, 2009*).

Our finding that αKD blocks lamellipodial and filopodial formation and cytoskeletal dynamics is also consistent with the WAVE clustering model. Taken together, we suggest that Pcdhα regulates the formation and dynamics of lamellipodial and filopodial protrusions underlying cortical neuron migration through the WAVE/Pyk2/Rac1 axis (*Figure 7*). We noted that αKD neurons stall in the lower intermediate zone and Pyk2OE neurons stall in the middle intermediate zone. In other words, αKD phenotype is more severe than that of Pyk2OE. In addition, αKD neurons display stunted processes while Pyk2OE neurons have branchy morphology. Consistently, the WAVE clustering model suggests that, in addition to disinhibition of Pyk2 and consequently inhibition of Rac1, αKD also impairs the membrane recruiting of the WAVE complex directly (*Figure 7*).

It is puzzling why Pcdhαc2 is different from other members of the Pcdhα family (*Figures 2A*, *5C and D*, and *Figure 2—figure supplement 1D*, *Figure 5—figure supplement 1C and D*). However, a recent study revealed an intriguing role of αc2 in serotonergic axonal local tiling and global assembly (*Chen et al., 2017*). Given the known role of variable cytoplasmic domain of clustered Pcdh proteins in their cytoplasmic association (*Shonubi et al., 2015*), the unique sequences of the αc2 variable cytoplasmic domain may restrict its role to axonal tiling of serotonergic neurons but not cortical neuron migration.

Diverse roles of the clustered *Pcdh* genes in axonal targeting, dendritic tiling and self-avoidance, spine morphogenesis, synaptogenesis and connectivity have been reported (*Garrett et al., 2012*; *Katori et al., 2009*; *Lefebvre et al., 2012*; *Molumby et al., 2016*; *Rubinstein et al., 2015*; *Suo et al., 2012*; *Thu et al., 2014*; *Zipursky and Sanes, 2010*). In particular, genetic studies demonstrated that Pcdhα functions in axonal projection of olfactory sensory and serotonergic neurons (*Chen et al., 2017*; *Hasegawa et al., 2008*; *Katori et al., 2009*; *Mountoufaris et al., 2017*). In addition, another WIRS-containing protocadherin, Celsr3, is also central for interneuron tangential migration and *Globus Pallidus* axonal connectivity in the mouse forebrain (*Jia et al., 2014*; *Ying et al., 2009*). It will be interesting to see whether these diverse protocadherin functions, in addition to the crucial role in cortical neuron migration, also require the complex WAVE/Pyk2/Rac1 signaling cascade (*Figure 7*). Sholl analysis demonstrated that the WIRS domain point mutation rescues the Pcdhα dominant-negative effects on dendrite outgrowth and branching of primary cultured cortical neurons, suggesting that the Pcdhα/WAVE/Pyk2/Rac1 signaling axis indeed functions in dendrite morphogenesis (*Figure 7—figure supplement 1*). Thus, the regulation of neuronal migration and neurite development by the Pcdhα/WAVE/Pyk2/Rac1 axis through actin cytoskeletal dynamics may be a general mechanism for diverse roles of protocadherins in brain development and function.

# Materials and methods

## Key resources table

| Reagent type (species) or resource | Designation | Source or reference | Identifiers | Additional information |
|---|---|---|---|---|
| Gene (*Mus musculus*) | *Pcdha6* | GenBank | GenBank: NM_007767.3 | N/A |
| Gene (*Mus musculus*) | *Pcdhac1* | GenBank | GenBank: NM_001003671.1 | N/A |
| Gene (*Mus musculus*) | *Pcdhac2* | GenBank | GenBank: NM_001003672.1 | N/A |
| Gene (*Mus musculus*) | *Ptk2b (Pyk2)* | GenBank | GenBank: NM_001162366.1 | N/A |
| Gene (*Mus musculus*) | *Wasf2 (WAVE2)* | GenBank | GenBank: AY135643.1 | N/A |
| Gene (*Mus musculus*) | *Abi2* | GenBank | GenBank: NM_198127.2 | N/A |
| Gene (*Mus musculus*) | *Rac1* | GenBank | GenBank: NM_009007.2 | N/A |
| Strain (*Mus musculus*) | Pcdhα$^{del/del}$ | doi:10.1038/ng2060 | N/A | N/A |
| Strain (*Mus musculus*) | Pcdhα$^{GFP}$ | doi:10.1038/ng2060 | N/A | N/A |
| Strain (*Mus musculus*) | Pyk2KO | DOI: https://doi.org/10.1101/216770 | N/A | N/A |
| Strain (*Mus musculus*) | Pyk2$^{Y402F}$ | DOI: https://doi.org/10.1101/216770 | N/A | N/A |
| Cell line (*Homo sapiens*) | HEK293T/17 | ATCC | Cat# CRL-11268 | N/A |
| Antibody | anti-beta-actin (mouse monoclonal) | Proteintech | Cat# 60009–1-Ig, RRID:AB_2687938 | N/A |
| Antibody | anti-Myc (mouse monoclonal) | Millipore | Cat# 05–724, RRID:AB_309938 | N/A |
| Antibody | anti-WAVE2 (rabbit polyclonal) | Millipore | Cat# 07–410, RRID:AB_310593 | N/A |
| Antibody | anti-Pyk2 (rabbit polyclonal) | Abcam | Cat# ab32571, RRID:AB_777566 | N/A |

*Continued on next page*

*Continued*

| Reagent type (species) or resource | Designation | Source or reference | Identifiers | Additional information |
|---|---|---|---|---|
| Antibody | anti-Tbr2 (rabbit polyclonal) | Abcam | Cat# ab23345, RRID:AB_778267 | N/A |
| Antibody | anti-Tuj1 (mouse monoclonal) | Covance | Cat# MMS-435P, RRID:AB_2313773 | N/A |
| Antibody | anti-Pcdhα (rabbit polyclonal) | Synaptic Systems | Cat# 190003 | N/A |
| Antibody | anti-GFP (rabbit polyclonal) | Invitrogen | Cat# A-31852, RRID:AB_162553 | N/A |
| Cat# | anti-BLBP (rabbit polyclonal) | Millipore | Cat# ABN14, RRID:AB_10000325 | N/A |
| Antibody | anti-GM130 (mouse monoclonal) | BD Bioscience | Cat# 610822, RRID:AB_398141 | N/A |
| Antibody | anti-BrdU (mouse monoclonal) | Bio-Rad | Cat# MCA2483, RRID:AB_808349 | N/A |
| Antibody | anti-activated caspase 3 (rabbit polyclonal) | Cell Signaling Technology | Cat# 9661, RRID:AB_2341188 | N/A |
| Software, algorithm | Prism | GraphPad (La Jolla, CA) | RRID:SCR_002798 | N/A |
| Software, algorithm | Fiji | doi: 10.1038/nmeth.2019 | RRID:SCR_002285 | N/A |
| Software, algorithm | Clustal X2 | doi: 10.1093/bioinformatics /btm404 | RRID:SCR_002909 | N/A |
| Recombinant DNA reagent | pCAG-EGFP (plasmid) | doi: 10.1523/JNEUROSCI .6096–09.2010 | N/A | N/A |
| Recombinant DNA reagent | pLKO.1-TRC cloning vector (plasmid) | Addgene | plasmid #10878 | N/A |
| Recombinant DNA reagent | pNeuroD-ires-GFP (plasmid) | doi: 10.1038/nn.2816 | N/A | N/A |
| Recombinant DNA reagent | pCAG-Pcdhα6 (plasmid) | This paper | N/A | vector: pCAG; cDNA fragment: mouse *Pcdha6* |
| Recombinant DNA reagent | pCAG-Pcdhαc1 (plasmid) | This paper | N/A | vector: pCAG; cDNA fragment: mouse *Pcdhac1* |
| Recombinant DNA reagent | pCAG-Pcdhαc2 (plasmid) | This paper | N/A | vector: pCAG; cDNA fragment: mouse *Pcdhac2* |
| Recombinant DNA reagent | pCAG-Pyk2 (plasmid) | This paper | N/A | vector: pCAG; cDNA fragment: mouse *Pyk2* |
| Recombinant DNA reagent | pCAG-WAVE2 (plasmid) | This paper | N/A | vector: pCAG; cDNA fragment: mouse *WAVE2* |
| Recombinant DNA reagent | pCAG-Abi2 (plasmid) | This paper | N/A | vector: pCAG; cDNA fragment: mouse *Abi2* |
| Recombinant DNA reagent | pCAG-Rac1 (plasmid) | This paper | N/A | vector: pCAG; cDNA fragment: mouse *Rac1* |
| Recombinant DNA reagent | pNeuroD-Pyk2-ires-GFP (plasmid) | This paper | N/A | vector: pNeuroD-ires-GFP; cDNA fragment: mouse *Pyk2* |
| Recombinant DNA reagent | pLKO.1-Pcdhα shRNA1 (plasmid) | This paper | N/A | vector: pLKO.1-TRC; target: aacagtatccagtgcaacacc |
| Recombinant DNA reagent | pLKO.1-Pcdhα shRNA2 (plasmid) | This paper | N/A | vector: pLKO.1-TRC; target: aattcattatcccaggatctc |

## Animals

The Pcdhα$^{GFP}$ mice were previously described (*Suo et al., 2012*; *Wu et al., 2007*). Pyk2KO and Pyk2$^{Y402F}$ mice were generated by CRISPR/Cas9. Animals were maintained at 23°C in a 12 hr (7:00–19:00) light and 12 hr (19:00–7:00) dark schedule. The day of vaginal plug was considered to be

embryonic day 0.5 (E0.5). All animal experiments were approved by the Institutional Animal Care and Use Committee (IACUC) of the Shanghai Jiao Tong University.

## Generation of CRISPR mice

Mouse lines of Pyk2KO and Pyk2$^{Y402F}$ were generated by using CRISPR/Cas9. Briefly, sgRNA scaffold sequences were constructed in the pLKO.1 plasmid. The construct was then used as template for amplifying a PCR product containing T7 promoter and sgRNA target sequence. The PCR product was gel-purified and used as templates for in vitro transcription of sgRNA (T7-Transcription Kit, Invitrogen). Cas9 mRNA was transcribed in vitro from linearized pcDNA3.1-Cas9 plasmid (T7-ULTRA-Transcription Kit, Ambion). Both Cas9 mRNA and sgRNAs were purified (Transcription Clean-Up Kit, Ambion), mixed in M2 (Millipore) at the concentration of 100 ng/μl, and then injected into the cytoplasm of fertilized eggs of C57BL/6 mice. For Pyk2$^{Y402F}$ mice, single-stranded oligo-donor nucleotides (ssODN) with mutation at Y402 residue and nonsense mutation at PAM sequence were co-injected together with the Cas9 mRNA and sgRNA. After equilibration for 30 min, 15–25 injected fertilized eggs were transferred into fallopian tube of pseudopregnant ICR mouse females. Offspring of these mice were genotyped by PCR, restriction endonuclease digestion, and Sanger sequencing. All oligos used are listed in *Supplementary file 1*.

## Antibodies

The following antibodies were used for biochemistry experiments: mouse anti-β-actin (1:5000, Proteintech), mouse anti-Myc (1:1000, Millipore), rabbit anti-Pyk2 (1:500, Abcam). The following antibodies were used for immunocytochemistry and immunohistochemistry: mouse anti-Tuj1 (1:300, Covance), rabbit anti-αCD (1:500, Synaptic Systems), rabbit anti-GFP (1:1000, Invitrogen), rabbit anti-BLBP (Brain lipid binding protein) (1:500, Chemicon), mouse anti-GM130 (1:1000, BD Bioscience), rat anti-BrdU (1:1000, Bio-Rad), rabbit anti-activated caspase 3 (1:500; Cell Signaling Technology), rabbit anti-Tbr2 (1:500, Abcam), rabbit anti-WAVE2 (1:500, Millipore), goat anti-rabbit Alexa Fluor 488 (1:300, Molecular Probes), goat anti-rabbit Alexa Fluor 568 (1:300, Molecular Probes), goat anti-mouse Alexa Fluor 568 (1:300, Molecular Probes), goat anti-mouse Alexa Fluor 647 (1:300, Molecular Probes).

## Plasmid construction

Full-length cDNAs of *Pcdha6*, *Pcdhac1*, *Pcdhac2*, *WAVE2* (GenBank AY135643.1), *Abi2* (GenBank NM_198127.2) were cloned from mouse brain total RNA preparations by reverse transcriptase PCR (RT-PCR). The cDNAs of *Myr-αCD*, *Rac1* and *Rac1$^{G12V}$*, *Pyk2* and *Pyk2* mutations (*Pyk2$^{Y402F}$*, *Pyk2$^{K457A}$*, *Pyk2$^{Y579F}$*, *Pyk2$^{Y580F}$*, *Pyk2$^{Y881F}$*), *Pyk2* fragments (ΔFERM, ΔFAT, FERM domain, Kinase domain) were cloned from previously published plasmids (*Suo et al., 2012*). WIRS-mutated and αKD-resistant *Pcdhα* isoforms (*α6\**, *αc1\**, *αc2\**, *Myr-αCD\**, *α6\*-AA*, *αc1\*-AA*, *Myr-αCD\*-AA*, *Myr-α6ICD\**, *Myr-αc1ICD\**, *Myr-αc2ICD\**), *Rac1$^{Q61L}$*, were constructed from the above plasmids. Constructs used in IUE for overexpression were cloned into the pCAG-Myc vector or pNeuroD-IRES-GFP vector (kindly provided by Dr. Franck Polleux, Columbia University) using restriction enzyme sites. For knockdown, short-hairpin RNA (shRNA) coding sequences were cloned into the pLKO.1 vector. All oligo sequences with corresponding restriction enzyme sites are listed in *Supplementary file 1*. Plasmids were validated by Sanger sequencing.

## In utero electroporation (IUE)

IUE was performed as previously described with modifications (*Saito and Nakatsuji, 2001*). Briefly, dams were anesthetized with pentobarbital sodium. pLKO.1-shRNAs (2 μg/μl) for knockdown or pCAG-Myc (2 μg/μl) constructs for overexpression were mixed with GFP-expressing plasmid pCAG-eGFP (0.5 μg/μl) and 0.05% fast green. Laparotomy was performed to expose the uteri. The plasmid mixture was injected into the lateral ventricle of the embryonic brain. Five electrical pulses were applied at 40 Volts for a duration of 50 ms at 900 ms intervals using a tweezertrode (3 mm, BTX) with an electroporator (Gene Pulser System, Bio-Rad). The uterine horns were placed back into the abdominal cavity to allow the embryos to continue normal development.

## Cortical neuron primary culture, organotypic slice culture, and time-lapse imaging

For cortical neuron primary culture, electroporated cortices were collected from E17.5 embryos in Hanks' Balanced Salt Solution (HBSS) with 0.5% glucose, 10 mM Hepes, 100 µg/ml penicillin/streptomycin. The cortices were then digested with 0.25% trypsin for 10 min at 37°C. The reaction was terminated with 0.5 mg/ml trypsin inhibitor for 3 min at room temperature (RT). The cortical tissues were gently triturated in the plating medium (MEM with 10% FBS, 1 mM glutamine, 10 mM Hepes, 50 µg/ml penicillin/streptomycin) until fully dissociated. Cell viability and density were determined using 0.4% trypan blue and a hemocytometer. The dissociated cells ($1 \times 10^5$) were plated into four-well chamber or 35-mm glass-bottom Petri dish precoated with 100 µg/ml poly-L-lysine (Sigma) and 5 µg/ml laminin (Invitrogen). The cells were incubated with 5% $CO_2$ at 37°C for 4 hr. The plating medium was then replaced with a serum-free culture medium (Neurobasal medium, 2% B27, 0.5 mM glutamine, 50 µg/ml penicillin/streptomycin supplemented with 25 µM glutamate). For immunocytochemistry, cells were cultured for additional 20 hr in vitro (hiv).

For cortical organotypic slice culture, the head of E17.5 embryos were briefly placed in 70% ethanol and the brains were carefully dissected. The brains were embedded in 3% low-melting agarose and glued to the chuck of a water-cooled vibratome (Leica). The 250-µm-thick whole-brain coronal sections were cut and collected in the sterile medium. The organotypic slices were carefully placed in a 0.4 µm membrane cell culture insert (Millipore) in a six-well plate. Slices were cultured in slice culture medium: 67% Basal Medium Eagle (BME), 25% HBSS, 5% FBS, 1% N2, 1% penicillin/streptomycin/glutamine (Invitrogen) and 0.66% glucose (Sigma). Slices (three per well) were cultured in six-well plates at 37°C and 5% $CO_2$, incubated for 6–8 hr. The membrane insert with slices was then transferred on to a glass-bottom Petri dish (MatTek). Images were taken at 3 µm steps with 10–15 optical sections and were captured every 15 min for up to 16 hr with the Nikon A1 confocal laser microscope system.

For single-cell time-lapse imaging, cortical neurons were plated into a 35-mm glass-bottom Petri dish. Images were taken at 1 µm steps with 10–15 optical sections and were captured every 5 min for up to 10 hr with Nikon A1 confocal laser microscope system.

## Immunocytochemistry, immunohistochemistry, and imaging

Primary cultured cortical neurons were washed once with PBS, fixed in 4% PFA for 20 min at RT, washed and permeabilized with 0.2% Triton X-100 for 10 min. After blocking with 5% BSA, cells were incubated with primary antibodies at 4°C overnight followed by incubation of secondary antibodies for 1–2 hr at RT. F-actin was labeled by Alexa-546 phalloidin (Sigma). For immunohistochemistry, the dams were sacrificed, and embryonic brains were fixed in 4% PFA overnight at 4°C. The brains were then sectioned at 50 µm with a vibratome (Leica). Sections were washed three times in PBS, blocked in 3% BSA, 0.1% Triton X-100 in PBS for 1 hr at RT, and then incubated with primary antibodies at 4°C overnight and secondary antibodies at RT for 1–2 hr. Cell nuclei were visualized with DAPI. Images were collected with a confocal microscope (Leica) under a 10x objective for brain sections. High-resolution images were collected under a 60x oil objective with a 3x digital zooming factor for primary cultured neurons.

## Cell culture and western blot

HEK293T cells were maintained in DMEM with 10% FBS and 100 µg/ml penicillin/streptomycin. Cultured cells were transfected using Lipofectamine 2000 (Invitrogen). Total protein of HEK293T cells was extracted by lysis buffer (50 mM Tris–HCl, pH 7.5, 150 mM NaCl, 1% NP40, 0.5% sodium deoxycholate, 0.1% SDS) with protease inhibitors and then centrifuged at 12,000 $\times$ g at 4°C for 30 min. The lysates were subjected to Western blot analyses.

## Reverse transcriptase PCR (RT-PCR)

Total RNA was extracted from embryonic mouse brain tissues with TRIzol (Ambion). The reverse-transcription reaction was performed with 1 µg total RNA preparations. All oligos used are listed in *Supplementary file 1*.

## Statistical analysis/image analysis and quantification

For each group, the *IUE* experiments were performed using at least three pregnant female mice, by which we usually harvested at least six embryonic brains. We obtained 15 ~ 20 sections from each electroporated brain, and quantified one typical section per brain. Nearly identical areas in the presumptive somatosensory cortices of anatomically matched brain sections were chosen for imaging and quantification. For bin analysis, the cortices were divided into ten equal bins and all $GFP^+$ neurons in each bin were counted. In total, about 150 ~ 300 cells were counted per section. Statistical significance was assessed using one-way ANOVA, followed by a post hoc Tukey's multiple comparisons test.

In primary culture experiments, the development stage of cultured neurons were defined as in Dotti's paper: at stage 1, the cell body was surrounded by flattened lamellipodia; at stage 2, the lamellipodia transformed into neural processes with growth cones (*Dotti et al., 1988*). We immunostained the cultured cells with Tuj1 (Neuron-specific class III beta-tubulin) antibody, a neuron-specific marker, to exclude differentiated glia or radial glia. For quantification, we selected neurons with typical stage 1 or stage 2 morphology based on GFP and phalloidin signals. For stage 1 neurons, we selected the lamellipodia region by the wand tool in the ImageJ software (NIH) and measured the area size. For stage 2 neurons, the neurite tips with F-actin-enriched protrusions two folds larger than its width were defined as 'neurite with lamellipodia'. Sholl analysis was performed as previously described (*Suo et al., 2012*).

The significance of differences between two groups was analyzed using unpaired Student's *t* tests. One-way ANOVA was used for multiple comparisons by the GraphPad software.

## Acknowledgements

We thank J Ao and Y Guo for technical help, and S Zhao for suggestions on live imaging. We thank WV Chen, T Maniatis, G Mountoufaris, T Südhof, X Wang, and L Wang, as well as members of the Wu Lab for critical reading of the manuscript. F Polleux for providing pNeuroD-IRES-EGFP plasmid. This study is supported by grants from NSFC (31630039, 91640118, and 31470820), the Ministry of Science and Technology of China (2017YFA0504203), the Science and Technology Commission of Shanghai Municipality (14JC1403601). QW is a Shanghai Subject Chief Scientist.

## Additional information

### Funding

| Funder | Grant reference number | Author |
| --- | --- | --- |
| National Natural Science Foundation of China | 31630039 | Qiang Wu |
| National Natural Science Foundation of China | 91640118 | Qiang Wu |
| National Natural Science Foundation of China | 31470820 | Qiang Wu |
| Ministry of Science and Technology of the People's Republic of China | 2017YFA0504203 | Qiang Wu |
| Science and Technology Commission of Shanghai Municipality | 14JC1403601 | Qiang Wu |

The funders had no role in study design, data collection, and interpretation, or the decision to submit the work for publication.

### Author contributions

Li Fan, Data curation, Formal analysis, Investigation, Writing—original draft; Yichao Lu, Data curation, Formal analysis, Investigation, Writing—original draft, Writing—review and editing; Xiulian

Shen, Hong Shao, Lun Suo, Data curation, Formal analysis, Investigation; Qiang Wu, Conceptualization, Supervision, Funding acquisition, Project administration, Writing—review and editing

### Author ORCIDs
Qiang Wu (iD) http://orcid.org/0000-0003-3841-3591

### Ethics
Animal experimentation: Animal experimentation: All procedures involving animals were in accordance with the Shanghai Municipal Guide for the care and use of Laboratory Animals, and approved by the Shanghai Jiao Tong University Animal Care and Use Committee (protocol #: 1602029).

### Decision letter and Author response
Decision letter https://doi.org/10.7554/eLife.35242.043
Author response https://doi.org/10.7554/eLife.35242.044

## Additional files

### Supplementary files
• Supplementary file 1. Oligonucleotides used in this study.
DOI: https://doi.org/10.7554/eLife.35242.033

• Transparent reporting form
DOI: https://doi.org/10.7554/eLife.35242.034

### Data availability
All data generated or analysed during this study are included in the manuscript and supporting files. Source data files have been provided for all figures.

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
