## [Decision Letter]

Thank you for submitting your manuscript "A Pcdhα/WRC/Pyk2/Rac1 Axis for Cortical Neuron Migration and Lamellipodial Dynamics" to *eLife*. Your article has been reviewed by a Senior Editor, a Reviewing Editor, and three reviewers. As you will see, all of the reviewers were impressed with the importance and novelty of your work.

In the reviews and the follow-up discussion, the major points regarding new experimental data that emerged are: (1) it would be useful to see comparison of the Pcdhα KO vs. the Pcdhα KD (knockdown) phenotypes; (2) likewise, a comparison between Pcdhα KD vs. Pyk2-OE (over-expression) phenotypes to determine the extent to which the defects are similar and thus consistent with misregulation of a Pcdhα-WRC-Pyk2 axis in neuronal migration and cytoskeletal dynamics; (3) assessing whether the mutant versions of Pcdhα go to the plasma membrane; and (4) describe sample sizes and consider other statistical tests such as ANOVA for comparing multiple groups.

We are including the three reviews at the end of this letter, as there are many specific and useful suggestions in them. We appreciate that the reviewers' comments cover a broad range of suggestions for improving the manuscript. Please use your best judgment in deciding which of these can be accommodated in a reasonable period of time. We look forward to receiving your revised manuscript.

Reviewer #1:

Cortical neurons born from the proliferative ventricular zone and subventricular zones migrate radially to reach the appropriate laminar positions during development. Wu et al. have investigated the role of protocadherin cell surface molecules, and their downstream signaling through the WAVE complex and Pyk2 kinase, in radial migration. The authors carried out an extensive set of experiments utilizing in vivo genetic manipulation, live imaging, immunohistochemistry, and primary cell culture to address this question. Using electroporation of shRNA targeting the common intracellular domain of the protocadherin-α cluster (Pcdhα), they find a dramatic effect on migration: Pcdhα knockdown sequesters neurons within the lower intermediate zone and inhibits them from arriving at the cortical plate. This effect is then shown to involve signaling through Pyk2, WAVE, and Rac1. Knockdown of Pcdhα elevates Pyk2 function; the deleterious effects of increased Pyk2 are shown to require Pyk2 kinase activity as well as its scaffolding FERM domain.

Overall, this is an interesting study. Little is known about signaling downstream of protocadherins, so this is a nice contribution. The authors provide a thorough and convincing picture of how Pcdhα/WRC/PKY2/Rac1 axis affects cortical neuron migration. The findings are generally well supported by the data presented. However, additional data may help make this study more impactful and is needed to support some of the conclusions.

Specific comments:

1) The authors demonstrated a clear lamellipodial phenotype in culture. However, its link to neuron phenotype in vivo was fuzzy. For one thing, the "stunted neurites" (subsection “Defective cortical neuron migration with Pcdhα knockdown”) morphology was not well documented. I had trouble appreciating the neuronal morphology in Figure 1E. The authors should include drawings of individual cells, as was done in Figure 3C (Pyk2 overexpression), because it is hard to tell the morphological detail in these panels.

2). Related to this: Based on the images provided in Figure 1E and Video 1 and Video 2, the Pyk2 overexpressing cells seem more branchy than Pcdhα knockdown cells. The authors' model requires both manipulations to affect the same underlying actin cytoskeletal biology. How do the authors explain the difference in morphology?

3) Last point on the morphology of migrating cells: It is unclear whether the "stunted neurite" morphology of Pcdhα-knockdown cells is due to altered lamellipodia. Same is true for extra branches in Pyk2 overexpressing cells. I thought I could discern some lamellipodia-like structures in Video 1 and Video 2, although without actin labeling it's hard to be sure. To connect the in vivo phenotype to the cultured cell phenotype, the authors should use an actin marker (e.g. lifeact) to find out if lamellipodia are affected in vivo.

4) The authors state that the WIRS motif is located in the common intracellular domain shared by most Pcdhα family members. They cite Chen et al., 2014 for this information. This is the paper that shows interaction of WAVE complex with the WIRS domain of Pcdhα6. I could not find documentation in that paper of where WIRS is located (i.e. is it in the variable or common cytoplasmic region). The authors should clarify how they determined that WIRS is in the common region. If it were in the variable region, they would need to show biochemical interactions with WAVE for other Pcdhα isoforms, to demonstrate that their point mutants are indeed disrupting this particular interaction. As long as it's in the common region I think the interaction shown in Chen et al. is sufficient.

5) There is not enough information provided on how the bin analysis was done to measure neuronal migration. The authors state "n=6." I assume that is the number of animals (please confirm). But how many sections were analyzed per animal, and how many cells were analyzed per section? How were particular sections/cells/fields of view selected for analysis? Also, since statistical comparisons were presumably made for multiple bins, the authors should justify the use of a T-test (rather than ANOVA with post-hoc comparisons) and/or explain how they controlled for multiple comparisons.

6) The authors concluded that Rac1Q61L cannot rescue the blocking activity of FERM domain alone. However, from the images shown in Figure 4—figure supplement 1D, this looks to me like a partial rescue compare to the control (FERM domain only) condition. In the absence of further data I don't necessarily agree with the authors' conclusion that FERM domain acts independently of Rac1. A related point: It would be helpful to have at least brief consideration in the Discussion of how the FERM domain could function without the kinase. This would help guide readers who don't know much about FERM domains.

7) The authors refer to cultured neurons at "Stage 1" and "Stage 2" but these terms are not defined. Please elaborate. Also, it should be stated in the methods how cultured cells were selected for analysis (e.g. is there a way to tell neurons from differentiated glia or radial glia). Finally, the Materials and methods section should include more detail on how lamellipodia were counted – i.e. what criteria were used to distinguish them from other protrusions.

8) There are many abbreviations throughout the manuscript. These abbreviations can make it hard for readers to follow the flow of the story. Since *eLife* has no length limits, the authors should endeavor to get rid of as many of these as is practical. Everything that is a two-letter abbreviation (e.g. IZ for intermediate zone) for sure, and hopefully others.

9) While the effects of downstream signaling from Pcdhα are demonstrated to be quite important, I was less clear on the authors' model for when this pathway might be activated. Do they think it is constitutively required? Or activated under certain circumstances – i.e. when Pcdhα mediates cell adhesion in a particular context? One possibility raised by the use of electroporation for the loss of function studies is that cells lacking Pcdhα might be at a relative disadvantage compared to the surrounding wild-type cells. This is a phenotype that has been observed for other cell adhesion molecules in other contexts. The authors have previously studied germline Pcdhα knockout mice; it would be quite informative if these animals had a phenotype that was less severe than the sparse knockdown.

10) Finally, a suggestion: Protocadherins have remarkably diverse extracellular domains that allow them to function in many cellular contexts. The authors argue that they are studying the common downstream output of this diverse protein family. It is a plausible argument based on the shared common intracellular domain. However, if they could show that the Pcdhα/WRC/PKY2/Rac1 axis generalizes to other Pcdhα functions – e.g. dendrite morphology, as they have previously studied, or self-avoidance – this would bolster their claim that they are in fact studying a common downstream pathway. Such a finding would, in my opinion, substantially increase the impact of the study. For example, the authors could test whether the WIRS domain point mutants can rescue Pcdhα loss-of-function effects on pyramidal cell dendrite branching.

For the title, all the abbreviations are probably going to stifle accessibility to a broad audience. "Protocadherin" and "WAVE complex" should substitute for their abbreviations, at minimum.

Reviewer #2:

Prior studies have demonstrated interaction between representative clustered and non-clustered protocadherins (Pcdhs) and the WAVE complex. Nevertheless, the biological relevance of Pcdh-VAVE interactions have not been demonstrated. In the current paper, Wu and colleagues report experiments in mice demonstrating that knockdown of α Pcdh leads to defects in neuronal migration, which can be rescued by reintroduction of a single a-Pcdh isoform. Mapping of the molecular regions responsible show that the cytoplasmic region can rescue alone, and this effect is lost by mutation of the a-Pcdh WAVE-binding WIRS sequence. Additional knockdown and overexpression experiments of non-receptor tyrosine kinase Pyk2, known to interact with Pcdh cytoplasmic regions, suggest that Pyk2 inhibits migration. Dissection of Pyk2 identifies the FERM domain as a key actor. Pyk2 appears to exert its effects by modulating the activation state of Rac1. Altogether, the authors define a putative pathway by which a-Pcdh interaction recruits the WAVE complex to the membrane to regulate neuronal migration. Concomitantly, a-Pcdh down-regulates Pyk2, leading to activation of WAVE through Rac1 activation.

Overall, this is a fascinating paper, which adds biological context to the previously reported interaction between Pcdhs and the WAVE complex. It should be of high interest to people in the field.

There are a number of questions raised by the findings, which are not addressed in the Discussion section:

1) How could ABI2 rescue if there's no anchor to the membrane?

2) Why is there no effect on brightness of GFP in αKD cells?

3) Two different Rac1 mutations have different effects, but potential interpretations are not discussed.

4) Overexpression of Pyk2 leads to a defect, which is further along than for αKD. Overexpression of the Pyk2 FERM domain recapitulates this phenotype, and overexpression of Pyk2 lacking the FERM domain appears wild-type. Since defects seem to be associated with FERM domain overexpression. It is hard to understand why the overexpression of a full-length kinase-dead Pyk2, containing the FERM domain, yields a wild-type phenotype. The authors should comment on this observation.

5) It would help to improve the description of the differences between the α-Pcdhs that give different phenotypes. These different phenotypes are presumably due to differences in the juxtamembrane "variable" cytoplasmic domain region. A sequence alignment showing differences between the α-Pcdhs would be useful. Also, it would help to mark the location of the WAVE-binding WIRS peptide.

Reviewer #3:

This is an interesting manuscript reporting on a pathway in which α-Pcdhs influence neuronal migration through regulation of WAVE-Pyk2-RAC1 signaling. The clustered Pcdhs regulate diverse aspects of neuronal patterning, but little is known of the pathways by which Pcdhs transduce signals and regulate cytoskeletal dynamics. This group and others have shown that Pcdhs influence neurite patterning by negatively regulating Pyk2 and FAK, and indirectly promoting activation of Rac1. Another group identified a WAVE-interacting sequence motif present in α-Pcdhs, suggesting that Pcdhαs may also signal through the WAVE complex (Chen et al., 2014).

Here, the authors test whether α-Pcdhs functionally interact with a WAVE-Pyk2-RAC1cascade by interrogating them in the context of cortical neuronal migration. They show that knock-down of Pcdhαs causes cortical migration defects, and then further manipulate Pcdhα and WAVE-Pyk2-RAC1 components through in utero electroporation and neuronal culture studies to determine their functional relationships. Their main conclusions are: (1) knockdown of Pcdhα specifically affects the extent and rate of migration, and this phenotype cannot be rescued by αc2 or by Pcdhα with mutations in the putative WAVE interacting motif (WIRS); (2) migration defects are observed when its downstream inhibitory target Pyk2 is overexpressed; and (3) the influence of α-Pcdhs on actin remodeling is also illustrated in lamellipodia formation and rescued by overexpression of Wave and Abl1 kinase.

Overall, the study is important and provides the first demonstration of functional interactions between WAVE and Pcdhαs. However, there are several concerns that need to be addressed (see below), which would require further experiments and analyses. The major weaknesses are the omission of studies using Pcdhα-KO tissue and the lack of biochemical data showing interactions between Pcdhα and WAVE components. Moreover, the study manipulates different components of the pathway, but it fails to compare the phenotypes to each other. And the manipulations on Pyk2, Rac1 etc., are not verified in the context of Pcdh-a KD-induced migration defects. Therefore, the studies fall short of demonstrating a Pcdhα-WAVE-Pyk2-Rac1 axis in the regulation of cortical migration and cytoskeletal dynamics.

1) The study is limited to shRNA-mediated knockdowns of Pcdhα, and does not extend to Pcdhα-KO brains. The omission is perplexing as the authors have generated and studied Pcdhα-KO mice in previous work (i.e. Wu et al., 2008; Suo et al., 2012). Data from Pcdhα-KO mice would significantly improve the quality of the results and, support the findings from the knockdown approaches. If they no longer carry these mice, they could obtain KO or conditional Pcdhα brains from other groups to report whether migration phenotypes are detected in fixed mutant tissue. If migration defects are not detected, they could further investigate if developmental delays, redundancy among Pcdhs, or possibly differential adhesion resulting from Pcdh mosaicism contributes to the mutant knockdown phenotype. Moreover, they could include an additional control for the shRNA by IUE Pcdhα-6 shRNA into Pcdh-aKO mice.

2) The images showing Pcdhα protein expression and co-localization with WAVE are not informative. In Figure 1, the panels are low magnification. High power images of Pcdhαs localized along processes of migrating neurons (in WT and IUE tissue with GFP+ labeled neurons) should be shown. In Figure 2, the localization of Pcdhα and WAVE appear to cytosolic rather than at the membranes. Have the authors detected their co-localization in neuronal membranes in migrating cells in IUE tissue, or in growth cones or lamellipodia (i.e. in Figure 5, and Figure 5—figure supplement 1; Xie et al., 2013), which would be relevant to this study? Note that other groups have used membrane-targeted GFP for IUE-mediated labeling of migrating neurons to better resolve neurite structures (i.e. Lyn-GFP in Xie et al., 2013).

3) The 'WIRS' sequence in Pcdhα6 proposed by Chen et al., (2014) resides in the 3rd constant exon, which is shared by all Pcdhαs, including αc2. Interestingly, the authors show that, in contrast to α6 and αc1, full-length αc2 does not rescue the phenotype. However, deleting the αc2 variableCD leads abolishes this effect, suggesting that the αc2 variableCD distinguishes the activity of αc2 from the other Pcdhαs. This is potentially important and could advance the idea Pcdhα isoforms have different functions through their VCDs. But follow-up is needed, especially given that Pcdhαs also differ in their extracellular domains. The authors could test if chimeric forms of full-length Pcdhα can rescue the migration phenotype (i.e. α6 ECD-αc2 VCD-αCD).

4) The finding in Figure 2G that mutating the WIRS motif in Pcdhαs fails to rescue migration is very interesting. However, it is premature to conclude: "Thus, Pcdhα regulates cortical neuron migration through the WRC complex". Additional data are needed: (1) Control experiments showing that this mutant Pcdhα-WIRS (AA) variant reaches the cell surface. There are reports in in vitro models suggesting that Pcdhαs do not traffic well to the cell-surface, and so it would be good to distinguish between alternate possibilities; and (2) Biochemical data showing interactions between Pcdhα and WAVE, such as pulldowns, preferably using cortical tissue. At the very least, pulldown assays in cell lines (as done by Chen et al., 2014) could be done to show that interactions are abolished with this mutant form.

5) In Figure 3, Pyk2OE also leads to neuronal migration phenotypes, but these mutant neurons are multipolar with increased branching, and their Golgi are misoriented. Were these phenotypes analyzed in Pcdhα-KD tissue? Migrating Pcdhα-KD neurons look bipolar in Figure 1, but there are no Lucida drawings. Likewise, does electroporating the constitutive Rac1 mutant rescue cortical defects in Pcdhα-KD? Again, comparing the same phenotypes across the manipulations would strengthen study's objective that a Pcdhα-WRC-Pyk2 axis regulates neuronal migration and cytoskeletal dynamics.

6) In subsection “Dissection of Pyk2 domain in cortical neuron migration”, the authors describe the Pyk2 structure-function analyses in terms 'recapitulating the migration defects of αKD'. 'Recapitulate' is misused here. Do they mean phenocopy? If that is the case, there is not sufficient evidence for this. As stated in point 5, the phenotypes were not evaluated in the same way, and Pyk2OE leads to multipolar phenotypes that is not shown for αKD. While the Pyk2 structure function analyses do reveal relevant domains, I fail to see how they inform on αKD regulation of Pyk2 activity. Moreover, these manipulations were limited to Pyk2, and were not coupled with αKD manipulations (I initially expected co-transfection experiments but found no description of this approach).

7) The migration defects are vaguely described in the Results section and are not sufficiently discussed in the context of previous studies. Many phenotypes are described, but the biological significances of these effects and whether the components produce the same effects are unclear.

For example, in subsection “A role of Pyk2 in cortical neuron migration”, the authors note that Pyk2OE leads to a multipolar phenotype. They interpret the results:

"Thus, Pyk2OE blocks multipolar migration by disrupting proper localization of the Golgi apparatus", but this statement is not fully supported by the data, nor are relevant citations given. Jossin et al., showed that Golgi orientation is important for orientating the direction of migration, but does not affect the speed. Here, Pyk2OE also affects migration speed.

Regarding this point: "Finally, early born Pyk2OE neurons are also stalled at IZ, suggesting that Pyk2 also plays a role in somal translocation (Figure 3—figure supplement 1C and 1D)."

The relevant example is presented in Figure 3H, in the live imaging. But no quantifications are presented.

Xie et al., 2013 showed that during the radial migration phase, cortical neurons undergo a multipolar-bipolar transition in their morphology for glia-guided locomotion, which is dependent on WAVE2, Abi2 (Xie et al., 2013). Is this relevant to the Pcdhα-WAVE-Pyk2 pathway? This could be expanded in the Discussion section.

8) The sequence of results are disconnected and the rationales are not clear.

For example, the transitions between Pcdhα to Pyk2 and back to Pcdhα are disconnected, and the two seem like separate studies.

Subsection “A role of Pyk2 in cortical neuron migration”

“We previously found that Pcdhα regulates dendritic and spine morphogenesis through inhibiting Pyk2 kinase activity (Suo et al., 2012). To this end, we investigated whether Pyk2 was involved in Pcdhs-regulated cortical neuron migration.”

The authors could better articulate their goal to test whether there is increased Pyk2 in αKD tissue, and provide experiments combining KD and Pyk2 manipulations.

Another example: It's not clear why the authors chose to further analyze the functional interactions between Pcdhα and Abi/Wave in lamellipodia using cultures of naïve, non-polarized neurons. Why not revisit this more mature lamellipodia structures relevant to migration, such as leading processes. The data on 'primary neurites' in Figure 5—figure supplement 1 are more convincing than those in Figure 5.

9) The method used for quantifying lamellipodia should be described. Which fluorescent structure was used? (GFP? F-Actin?). The GFP appears to be cytosolic and may not fully resolve the membrane structures.

10) Samples sizes are not properly described. Ns are given, but it's not clear what they comprise. The authors should report the numbers of cells, sections analyzed, animals per litter, numbers of litter/replicates.

11) Statistical tests are limited to t-tests. But in many instances, multiple groups are analyzed and thus ANOVA and post-hoc multiple comparisons would be more suitable.

12) I think it is also premature to exclude effects on survival, since quantification of numbers of GFP labeled cells in each IUE manipulation are not given. Cleaved-caspase 3 might only be detected during a small window, and labeled cells are cleared rapidly.

---

## [Author Response]

In the reviews and the follow-up discussion, the major points regarding new experimental data that emerged are: (1) it would be useful to see comparison of the Pcdhα KO vs. the Pcdhα KD (knockdown) phenotypes;

Thanks for your insightful synthesis. We have performed the experiments suggested by the three reviewers. The resulting new data are presented as new figures or figure panels. The new Figure1—figure supplement 1I shows new data of the Pcdhα knockout mice by IUE experiments with GFP plasmids. Quantification revealed that there is no significant difference between heterozygous and homozygous Pcdhα KO mice. Thus, there appears to be no cortical neuron migration defect in embryonic Pcdhα KO mice. This is consistent with the normal cortical layering of the adult Pcdhα KO mice reported in our original paper (Wu et al., 2007). In addition, there are not uncommon for the different phenotypes between acute gene inactivation by RNAi and constitutive germline knockout by gene targeting (Bai et al., 2003; Corbo et al., 2002; de Nijs et al., 2009; Koizumi et al., 2006; Pramparo et al., 2010; Rossi et al., 2015; Suzuki et al., 2009; Young-Pearse et al., 2007).

Perhaps the most famous and relevant examples are the established role of *doublecortin* and *doublecortin-like* in cortical neuron migration. Knockdown of either *doublecortin* or *doublecortin-like* results in clear and convincing cortical neuron migration defects. By contrast, knockout of either *doublecortin* or *doublecortin-like* results in no cortical neuron migration defects (Bai et al., 2003; Corbo et al., 2002; Gotz, 2003; Koizumi et al., 2006; Pramparo et al., 2010). In our case of the Pcdhα, we have two independent shRNA targeting different subregions of the common constant region of all members of the Pcdhα family. In addition, we also have clear rescue experiments by shRNA-resistant constructs. These data provide strong and convincing evidence for a role of Pcdhα in cortical neuron migration. We have added the following sentence to the Results section: “Finally, there is no cortical migration defect (Figure 1—figure supplement 1I) in mice with deletion of the entire Pcdhα cluster (αKO) (Wu et al., 2007). The phenotypic discrepancy may be due to known genetic compensation mechanisms induced by deletion but not knockdown (Rossi et al., 2015).”

(2) likewise, a comparison between Pcdhα KD vs. Pyk2-OE (over-expression) phenotypes to determine the extent to which the defects are similar and thus consistent with misregulation of a Pcdhα-WRC-Pyk2 axis in neuronal migration and cytoskeletal dynamics;

We have performed extensive in-vitro and in-vivo (with Lifeact) experiments to compare αKD vs. Pyk2OE phenotypes in cytoskeletal dynamics and neuronal migration. The resulting data are presented in the new Figure 6 and Figure 6—figure supplement 1. We renamed the old Figure 6 as the new Figure 7 accordingly. We added a new subsection to the Results section: “A comparison between PcdhαKD and Pyk2OE in cytoskeletal dynamic”:

“Consistent with that Pyk2KD rescues cortical neuron migration defects of PcdhαKD (Figure 3A), we found that knockdown of Pyk2 in αKD cells results in a significant increase of lamellipodial sizes of stage1 neurons as well as of the percentage of primary neurites with lamellipodia of stage2 neurons (Figure 6A-6C). In addition, Pyk2OE results in a significant decrease of lamellipodial sizes, consistent with that of αKD (Figure 6D and 6E). […] In the upper intermediate zone, Pyk2OE neurons exhibit branchy morphology with multiple aberrant processes; however, the control neurons have normal bipolar morphology with single leading processes and growth cones (Figure 6-figure supplement 1B).”

Although both α knockdown and Pyk2 overexpression result in cortical neuron migration defects, they do display subtle differences. We added the following two sentences to the Discussion section: “We noted that αKD neurons stall in the lower intermediate zone and Pyk2OE neurons stall in the middle intermediate zone. In other words, αKD phenotype is more severe than that of Pyk2OE. In addition, αKD neurons display stunted processes while Pyk2OE neurons have branchy morphology.”

(3) assessing whether the mutant versions of Pcdhα go to the plasma membrane;

Indeed, these control experiments are important as previous transfection experiments have suggested that cell-surface delivery of Pcdhα to neuronal membrane requires Pcdhγ (Bonn et al., 2007; Goodman et al., 2017; Murata et al., 2004; Schreiner and Weiner, 2010; Thu et al., 2014). To this end, we constructed C-terminal Myc-tagged mutant versions of α6*-AA, αc1*-AA, Myr-αCD*-AA, as well as their corresponding wildtypes α6*, αc1*, Myr-αCD*, and transfected each mutant version or their corresponding wildtype into primary cultured neurons, which are most likely expressing Pcdhγ. We then immunostained these primary cultured neurons with Myc antibody to show the location of transfected Pcdhα. Both wildtype and WIRS motif mutant versions are detected at the very tip of neurite ends, so we concluded that they go to the cell-surface. We presented the new data as Figure 2—figure supplement 1F and added the following sentence to the Results section: “As controls, these WIRS mutated isoforms as well as wildtypes appears to reach the plasma membrane (Figure 2—figure supplement 1F).”

and (4) describe sample sizes and consider other statistical tests such as ANOVA for comparing multiple groups.

We have gone through the relevant legends and methods to make corrections. In addition, we have described sample sizes and used ANOVA for comparing multiple groups. Finally, we have added the following subsection of detailed description to the Materials and methods section: “Statistical analysis/ Image analysis and quantification”

Reviewer #1:[…] Overall, this is an interesting study. Little is known about signaling downstream of protocadherins, so this is a nice contribution. The authors provide a thorough and convincing picture of how Pcdhα/WRC/PKY2/Rac1 axis affects cortical neuron migration. The findings are generally well supported by the data presented. However, additional data may help make this study more impactful and is needed to support some of the conclusions.

Indeed, the novelty of Pcdhα in cortical neuron migration supported by thorough and convincing evidence is the main finding of the work. We have performed additional experiments suggested by Reviewer #1 and obtained new data to make the study more impactful and to support some of the conclusions.

Specific comments:

*1) The authors demonstrated a clear lamellipodial phenotype in culture. However, its link to neuron phenotype* in vivo *was fuzzy. For one thing, the "stunted neurites" (subsection “Defective cortical neuron migration with Pcdhα knockdown”) morphology was not well documented. I had trouble appreciating the neuronal morphology in Figure 1E. The authors should include drawings of individual cells, as was done in Figure 3C (Pyk2 overexpression), because it is hard to tell the morphological detail in these panels.*

We thank Reviewer #1 for this suggestion and we have added Lucida drawings of typical cells as in new Figure 1E, to show the morphological differences between control and Pcdhα knockdown neurons. We have changed the sentence from: “The αKD multipolar neurons in IZ display stunted neurites (Figure 1E)” to: “The αKD multipolar neurons in IZ display stunted processes, as shown by lucida drawings (Figure 1E)”.

2) Related to this: Based on the images provided in Figure 1E and Video 1 and Video 2, the Pyk2 overexpressing cells seem more branchy than Pcdhα knockdown cells. The authors' model requires both manipulations to affect the same underlying actin cytoskeletal biology. How do the authors explain the difference in morphology?

We agree that Pyk2OE and αKD neurons have different morphology. As stated above, we have performed additional experiments to compare them and the resulting data are presented in new Figure 6. Our new data show that Pyk2OE results in a significant increase of filopodial number per stage1 neuron and of primary neurite number per stage2 neuron, consistent with more branchy phenotype. Pyk2OE leads to the inhibition of Rac1 activity (Suo et al., 2012). As Rac1 is thought to provide the spatial information for actin polymerization (Tahirovic et al., 2010), loss of Rac1 activity leads to aberrant actin polymerization at many sites with no controlled spatial information, resulting in more aberrant filopodia. This explains that immature Pyk2OE neurons display more branching while wildtype neurons are bipolar.

We therefore added the following sentence to the Results section: “Pyk2OE leads to the inhibition of Rac1 activity (Suo et al., 2012). As Rac1 is thought to provide the spatial information for actin polymerization (Tahirovic et al., 2010), loss of Rac1 activity leads to aberrant actin polymerization at many sites with no controlled spatial information, resulting in more branchy phenotype.”

*3) Last point on the morphology of migrating cells: It is unclear whether the "stunted neurite" morphology of Pcdhα-knockdown cells is due to altered lamellipodia. Same is true for extra branches in Pyk2 overexpressing cells. I thought I could discern some lamellipodia-like structures in Video 1 and Video 2, although without actin labeling it's hard to be sure. To connect the* in vivo *phenotype to the cultured cell phenotype, the authors should use an actin marker (e.g. lifeact) to find out if lamellipodia are affected* in vivo.

We thank Reviewer #1 for this insightful suggestion. To address this question, we performed the suggested experiment using the Lifeact to see whether lamellipodia are affected in αKD and Pyk2OE cells. Lifeact was constructed into pCAG plasmid with C-terminal mCherry in-frame fusion, so the Lifeact labeled actin can be observed by red fluoresce. We added a paragraph at the end of the Results section: “To see whether growth cones with lamellipodia and filopodia are affected in vivo, we co-electroporated Lifeact, an actin marker, with either αKD or Pyk2OE plasmids into the developing mouse cortex. […] In the upper intermediate zone, Pyk2OE neurons exhibit branchy morphology with multiple aberrant processes; however, the control neurons have normal bipolar morphology with single leading processes and growth cones (Figure 6—figure supplement 1B).”

4) The authors state that the WIRS motif is located in the common intracellular domain shared by most Pcdhα family members. They cite Chen et al., 2014 for this information. This is the paper that shows interaction of WAVE complex with the WIRS domain of Pcdhα6. I could not find documentation in that paper of where WIRS is located (i.e. is it in the variable or common cytoplasmic region). The authors should clarify how they determined that WIRS is in the common region. If it were in the variable region, they would need to show biochemical interactions with WAVE for other Pcdhα isoforms, to demonstrate that their point mutants are indeed disrupting this particular interaction. As long as it's in the common region I think the interaction shown in Chen et al. is sufficient.

We appreciate these comments and apologize for the confusion. As seen in the new sequence alignments in Author response image 1, the WIRS motif is in the common cytoplasmic region of all members of the Pcdhα protein family. Indeed, in the pioneering study by Chen et al., they reported the physical interactions between the WIRS motif of Pcdhα and the WRC (WAVE) complex (Chen et al., 2014).

**Author response image 1. respfig1:** The WIRS motif is in the common cytoplasmic constant domain (CD) of Pcdhα proteins.

5) There is not enough information provided on how the bin analysis was done to measure neuronal migration. The authors state "n=6." I assume that is the number of animals (please confirm). But how many sections were analyzed per animal, and how many cells were analyzed per section? How were particular sections/cells/fields of view selected for analysis? Also, since statistical comparisons were presumably made for multiple bins, the authors should justify the use of a T-test (rather than ANOVA with post-hoc comparisons) and/or explain how they controlled for multiple comparisons.

We have added a paragraph to provide detailed information on how the bin analysis was done to measure neuronal migration: “For each group, the IUE experiments were performed using at least three pregnant female mice, by which we usually harvested at least 6 embryonic brains. We obtained 15~20 sections from each electroporated brain and quantified 1 typical section per brain. Nearly identical areas in the presumptive somatosensory cortices of anatomically matched brain sections were chosen for imaging and quantification. For bin analysis, the cortices were divided into ten equal bins and all GFP+ neurons in each bin were counted. In total, about 150~300 cells were counted per section. Statistical significance was assessed using one-way ANOVA, followed by a post hoc Tukey’s multiple comparisons test.”

We thank Reviewer #1 for pointing out that some of the statistical analyses in manuscript are not correctly used. We agree that Student’s *t* tests are not appropriate for multiple comparisons. We re-performed all the significance tests using ANOVA with post-hoc comparisons. We added one sentence to the end of the above paragraph.

6) The authors concluded that Rac1Q61L cannot rescue the blocking activity of FERM domain alone. However, from the images shown in Figure 4—figure supplement 1D, this looks to me like a partial rescue compare to the control (FERM domain only) condition. In the absence of further data I don't necessarily agree with the authors' conclusion that FERM domain acts independently of Rac1. A related point: It would be helpful to have at least brief consideration in the Discussion of how the FERM domain could function without the kinase. This would help guide readers who don't know much about FERM domains.

We thank Reviewer#1 for this comment. Because there appears to be more neurons migrated into the cortical plate than FERM in this particular section, we agree that it looks like partial rescue of FERM+Rac1^Q61L^ in old Figure 4—figure supplement 1D. However, as shown in (Author response figure 2) of all six sections from six mouse brains, overall there does not appear to be significant different between FERM+Rac1^Q61L^ and FERM alone. We have nevertheless replaced the FERM+Rac1^Q61L^ image in old Figure 4—figure supplement 1D with a new one (Figure 4—figure supplement 1D) to avoid potential confusion.

It was previously reported that the FERM domain of Pyk2 has a role in malignant glioma cell migration (Lipinski et al., 2006), and Pyk2 FERM domain is involved in the regulation of Pyk2 activity by acting to regulate the formation of Pyk2 oligomers which is critical for Pyk2 activity (Riggs et al., 2011). Overwhelming evidence in literature shows that the FERM domain of FAK, a close homolog of Pyk2, has clear kinase-independent scaffolding activity by interacting with numerous partners and regulating downstream signaling (Frame et al., 2010). As suggested we have added the following sentence to the Discussion section to help guide readers about the FERM domain scaffolding: “Moreover, our data suggest that Pyk2 also has kinase-independent scaffolding activity through its FERM (four-point-one, ezrin, radixin, moesin) domain, similar to the FERM domain of FAK, which binds to numerous interacting partners and connects cell cortex to diverse downstream intracellular pathways (Frame et al., 2010).

**Author response image 2. respfig2:** Rac1^Q61L^ cannot rescue the blocking activity of FERM domain.

7) The authors refer to cultured neurons at "Stage 1" and "Stage 2" but these terms are not defined. Please elaborate. Also, it should be stated in the methods how cultured cells were selected for analysis (e.g. is there a way to tell neurons from differentiated glia or radial glia). Finally, the Materials and methods section should include more detail on how lamellipodia were counted – i.e. what criteria were used to distinguish them from other protrusions.

Thanks for the suggestion. We added the following to subsection “Pcdh in lamellipodial formation and cytoskeletal dynamics”: “The early development of primary cultured neurons can be divided into two stages: stage1, in which the cell body is surrounded by flattened lamellipodia, and stage2, in which the lamellipodia transform into definitive processes with growth cones (Dotti et al., 1988).”

We selected neurons with typical stage1 or stage2 morphology based on the GFP signal, which labels transfected cells, and the phalloidin signal, which stains F-actin.

We immunostained the cultured cells with a Tuj1 (Neuron-specific class III β-tubulin) antibody, which is regarded as a neuron-specific marker, to exclude differentiated glia or radial glia.

For stage1 neurons, we selected the lamellipodia region by the wand tool in the ImageJ software, and measured the area’s size. For stage2 neurons, the neurite tips with F-actin-enriched protrusions two folds larger than the neurite width are defined as neurites with lamellipodia.

8) There are many abbreviations throughout the manuscript. These abbreviations can make it hard for readers to follow the flow of the story. Since eLife has no length limits, the authors should endeavor to get rid of as many of these as is practical. Everything that is a two-letter abbreviation (e.g. IZ for intermediate zone) for sure, and hopefully others.

Thanks for the suggestion and we have gone through the entire text carefully to get rid of as many as abbreviations as is practical.

9) While the effects of downstream signaling from Pcdhα are demonstrated to be quite important, I was less clear on the authors' model for when this pathway might be activated. Do they think it is constitutively required? Or activated under certain circumstances – i.e. when Pcdhα mediates cell adhesion in a particular context? One possibility raised by the use of electroporation for the loss of function studies is that cells lacking Pcdhα might be at a relative disadvantage compared to the surrounding wild-type cells. This is a phenotype that has been observed for other cell adhesion molecules in other contexts. The authors have previously studied germline Pcdhα knockout mice; it would be quite informative if these animals had a phenotype that was less severe than the sparse knockdown.

Cell motility, in particular cortical neuron migration, is enormously complex (Ayala et al., 2007; Krause and Gautreau, 2014; Mitra et al., 2005). The two highly-similar mammalian cell-adhesion kinases, Pyk2 and FAK, are likely central for cell motility. Our data suggest that Pcdhα and Pyk2 function in cortical neuron migration through WAVE complex to regulate actin fiber dynamics. We think that our working model is dynamic, undulating between activated and inactivated states through coordinated cycles in the cell leading edge and trailing edge. We do not know whether these activities are dependent on the cell-adhesion activity of Pcdhα or not. Nevertheless, in conjunction with the germline knockout experiments, our data suggest that cells lacking Pcdhα might be indeed at a relative disadvantage compared to the surrounding wild-type cells.

As stated above, we have performed the α germline knockout experiments and the new data are presented in Figure 1—figure supplement 1I. Indeed, the knockout mice appear to have no defect in cortical neuron migration. This is strikingly similar to members of the DOUBLECORTIN protein family as discussed extensively above.

10) Finally, a suggestion: Protocadherins have remarkably diverse extracellular domains that allow them to function in many cellular contexts. The authors argue that they are studying the common downstream output of this diverse protein family. It is a plausible argument based on the shared common intracellular domain. However, if they could show that the Pcdhα/WRC/PKY2/Rac1 axis generalizes to other Pcdhα functions – e.g. dendrite morphology, as they have previously studied, or self-avoidance – this would bolster their claim that they are in fact studying a common downstream pathway. Such a finding would, in my opinion, substantially increase the impact of the study. For example, the authors could test whether the WIRS domain point mutants can rescue Pcdhα loss-of-function effects on pyramidal cell dendrite branching.

We thank Reviewer #1 for this good suggestion and we have performed suggested dendritic experiment and Sholl analysis to address whether Pcdhα WIRS motif play a general role in dendrite development and branching. We previously showed that Myr-αCD function as a dominant-negative construct and lead to dendrite developmental defect in cultured hippocampal neurons (Suo et al., 2012). Our new experiments with primary cultured cortical neurons using Myr-αCD reproduced the dendrite developmental defect (Figure 7—figure supplement 1). Strikingly, mutation of the WIRS (from FITFGK to FIAAGK) abolished the dendrite developmental defect, suggesting that other Pcdhα functions such as dendrite development are also dependent on the signaling through the WAVE complex (Figure 7—figure supplement 1). We added the following sentence to the Discussion section: “Sholl analysis demonstrated that the WIRS domain point mutation can rescue the Pcdhα dominant-negative effects on dendrite outgrowth and branching of primary cultured cortical neurons, suggesting that the Pcdhα/WAVE/Pyk2/Rac1 signaling axis indeed functions in dendrite morphogenesis (Figure 7—figure supplement 1).”

For the title, all the abbreviations are probably going to stifle accessibility to a broad audience. "Protocadherin" and "WAVE complex" should substitute for their abbreviations, at minimum.

We have considered carefully and changed the title to: “Αlpha Protocadherins and Pyk2 kinase Regulate Cortical Neuron Migration and Cytoskeletal Dynamics via Rac1 GTPase and WAVE complex”.

Reviewer #2:[…] Overall, this is a fascinating paper, which adds biological context to the previously reported interaction between Pcdhs and the WAVE complex. It should be of high interest to people in the field.

Indeed, this work provides strong evidence for novel function of the previously reported interactions between Pcdhs and the WAVE complex in the brain.

There are a number of questions raised by the findings, which are not addressed in the Discussion section:1) How could ABI2 rescue if there's no anchor to the membrane?

We thank Reviewer#2 for this insightful question. There are four PXXP motifs within the common Pcdhα cytoplasmic constant domains which are anchors for ABI2 to the membrane. We have added the following two sentences to the Discussion section: “The WIRS motif of members of the Pcdhα family binds to a composite surface formed by Abi2 and Sra1 of WAVE (Chen et al., 2014). In addition, the Pcdhα proteins may also recruit WAVE through the direct binding of Abi2 C-terminal SH3 domain to the four PXXP motifs, which are specific to the constant domain of the Pcdhα but not Pcdhγ family (Wu and Maniatis, 1999).”

2) Why is there no effect on brightness of GFP in αKD cells?

We are sorry for the confusion. For all the α knockdown *IUE* experiments, we co-electroporated two separate plasmids: one plasmid expresses shRNA and the other one expresses GFP. The two plasmids should not influence each other. Thus, the GFP expression level is not decreased in αKD neurons.

The GFP is used to label transfected cells in the IUE experiments. If two plasmids are co-electroporated into the mouse brain by IUE, almost all of the transfected cells are co-transfected. This is proved by GFP-mCherry co-electroporation experiments (Author response image 3).

**Author response image 3. respfig3:** GFP and mCherry co-electroporation.

3) Two different Rac1 mutations have different effects, but potential interpretations are not discussed.

The two Rac1 mutations are likely have different constitutive activities because the Q61L mutant has a higher affinity for GTP than the G12V mutant (Heasman and Ridley, 2008; Luo et al., 1996; Xu et al., 1997). We have added the following at the end of the relevant sentence: “likely because it has a lower affinity for GTP and thus lower constitutive activity than Rac1^Q61L^.”

4) Overexpression of Pyk2 leads to a defect, which is further along than for αKD. Overexpression of the Pyk2 FERM domain recapitulates this phenotype, and overexpression of Pyk2 lacking the FERM domain appears wild-type. Since defects seem to be associated with FERM domain overexpression. It is hard to understand why the overexpression of a full-length kinase-dead Pyk2, containing the FERM domain, yields a wild-type phenotype. The authors should comment on this observation.

We thank Reviewer #2 for the comment. As discussed above, Pyk2 has both tyrosine kinase signaling and FERM domain scaffolding activity under physiological conditions. According to the structure of FAK (Lietha et al., 2007), which is a very close homolog of Pyk2 and likely shares similar structures, the FERM and kinase domains are sequestered in an autophosphorylation-closed inactive state. With the kinase-dead mutation, there is no conformational change for opening and phosphorylation, and subsequent Pyk2 activation. Therefore, it is understandable that the overexpression of a full-length kinase-dead Pyk2, despite containing the FERM domain, because of its closed state, still yields a wild-type phenotype of cortical neuron migration.

5) It would help to improve the description of the differences between the α-Pcdhs that give different phenotypes. These different phenotypes are presumably due to differences in the juxtamembrane "variable" cytoplasmic domain region. A sequence alignment showing differences between the α-Pcdhs would be useful. Also, it would help to mark the location of the WAVE-binding WIRS peptide.

We appreciate Reviewer #2 for this good suggestion. We performed the suggested sequence alignment to demonstrate the differences in the juxtamembrane "variable" cytoplasmic domain region (Figure 2—figure supplement 1E). Pcdhα1-Pcdhα12 have relatively short VCDs with high similarity, while Pcdhαc1 and Pcdhαc2 have longer and divergent VCDs. These structural differences may underlie the functional differences between diverse Pcdhα isoforms. We have added the following sentence to the Results section: “Consistently, sequence analysis revealed that αc2 VCD is the longest and most divergent among those of αc1 as well as of α1-α12 (Figure 2—figure supplement 1E).”

The WIRS motif is in the common cytoplasmic constant region of the Pcdhα proteins as described above in addressing reviewer #1’s comments (Author response image 1).

Reviewer #3:[…] 1) The study is limited to shRNA-mediated knockdowns of Pcdhα and does not extend to Pcdhα-KO brains. The omission is perplexing as the authors have generated and studied Pcdhα-KO mice in previous work (i.e. Wu et al., 2008; Suo et al., 2012). Data from Pcdhα-KO mice would significantly improve the quality of the results and, support the findings from the knockdown approaches. If they no longer carry these mice, they could obtain KO or conditional Pcdhα brains from other groups to report whether migration phenotypes are detected in fixed mutant tissue. If migration defects are not detected, they could further investigate if developmental delays, redundancy among Pcdhs, or possibly differential adhesion resulting from Pcdh mosaicism contributes to the mutant knockdown phenotype. Moreover, they could include an additional control for the shRNA by IUE Pcdhα-6 shRNA into Pcdh-aKO mice.

We appreciate Reviewer #3 for these comprehensive constructive suggestions. As described above, we have performed experiments with Pcdhα knockout mice and there appears no cortical neuron migration defect. We discussed the reasons extensively above. Because of limited time to revise the manuscript, we have not investigated the reasons experimentally. We appreciate the additional control experiment for the shRNA by IUE Pcdh α6 shRNA into α knockout mice. However, we have used two independent shRNAs targeting distinct subregions of the Pcdhα constant region. Most importantly, we have solid rescue results. The consilience of all of these results demonstrates that the αKD defect is not due to off-target.

2) The images showing Pcdhα protein expression and co-localization with WAVE are not informative. In Figure 1, the panels are low magnification. High power images of Pcdhαs localized along processes of migrating neurons (in WT and IUE tissue with GFP+ labeled neurons) should be shown. In Figure 2, the localization of Pcdhα and WAVE appear to cytosolic rather than at the membranes. Have the authors detected their co-localization in neuronal membranes in migrating cells in IUE tissue, or in growth cones or lamellipodia (i.e. in Figure 5, and Figure 5—figure supplement 1; Xie et al., 2013), which would be relevant to this study? Note that other groups have used membrane-targeted GFP for IUE-mediated labeling of migrating neurons to better resolve neurite structures (i.e. Lyn-GFP in Xie et al., 2013).

We thank Reviewer #3 for this comment. It’s hard to detect endogenous Pcdhα localization along processes of migrating neurons in WT and *IUE* tissues because there are too many cells compact together. The WAVE2 and Abi2 proteins co-localize with F-actin in lamellipodia enriched growth cones (Xie et al., 2013). We have performed GFP staining using the Pcdhα^GFP^ mice. As shown in Author response image 4, the endogenous Pcdhα proteins co-localize with F-actin in the growth cones. Together, we conclude that endogenous Pcdhα proteins co-localize with WAVE2/Abi2.

**Author response image 4. respfig4:** The localization of endogenous Pcdhα proteins with F-actin in growth cones.

3) The 'WIRS' sequence in Pcdhα6 proposed by Chen et al., (2014) resides in the 3rd constant exon, which is shared by all Pcdhαs, including αc2. Interestingly, the authors show that, in contrast to α6 and αc1, full-length αc2 does not rescue the phenotype. However, deleting the αc2 variableCD leads abolishes this effect, suggesting that the αc2 variableCD distinguishes the activity of αc2 from the other Pcdhαs. This is potentially important and could advance the idea Pcdhα isoforms have different functions through their VCDs. But follow-up is needed, especially given that Pcdhαs also differ in their extracellular domains. The authors could test if chimeric forms of full-length Pcdhα can rescue the migration phenotype (i.e. α6 ECD-αc2 VCD-αCD).

As can be seen from the VCD sequence alignment, each member of the Pcdhα family proteins has a unique VCD sequences, with the αc2 having the longest and most divergent VCD sequences. To test the VCD function, we performed experiments to delete the VCD domain of the α6, αc1, and αc2 protein (Author response image 5). Because myristoylated α constant domain as well as myristoylated α6 and αc1 intracellular domains (ICD) rescues α knockdown phenotype, it is intriguing that neither α6ΔVCD* nor αc1ΔVCD* rescues the αKD cortical neuron migration defect (Author response figure 5). These new data demonstrated again that αVCD is very important (Author response figure 5). Since we already showed the data on Myristoylated αc2 ICD and Myristoylated αc2 CD, namely the deletion of αc2 VCD, it is expected that the chimeric forms of full-length Pcdhα (i.e. α6 ECD-αc2 VCD-αCD) cannot rescue the migration phenotype. To emphasize the difference of αc2 VCD, as stated above, we have added one sentence near the end of the section2 of Results section: “Consistently, sequence analysis revealed that αc2 VCD is the longest and most divergent among those of αc1 as well as of α1-α12 (Figure 2——figure supplement 1E).”

**Author response image 5. respfig5:** Deletion of Pcdhα VCDs results in the abolishment of rescue of αKD defects.

*4) The finding in Figure 2G that mutating the WIRS motif in Pcdhαs fails to rescue migration is very interesting. However, it is premature to conclude: "Thus, Pcdhα regulates cortical neuron migration through the WRC complex". Additional data are needed: (1) Control experiments showing that this mutant Pcdhα-WIRS (AA) variant reaches the cell surface. There are reports in* in vitro *models suggesting that Pcdhαs do not traffic well to the cell-surface, and so it would be good to distinguish between alternate possibilities; and (2) Biochemical data showing interactions between Pcdhα and WAVE, such as pulldowns, preferably using cortical tissue. At the very least, pulldown assays in cell lines (as done by Chen et al., 2014) could be done to show that interactions are abolished with this mutant form.*

This has been addressed above. We have not done the pulldown because it will not provide stronger evidence than the Chen et al., 2014 paper.

5) In Figure 3, Pyk2OE also leads to neuronal migration phenotypes, but these mutant neurons are multipolar with increased branching, and their Golgi are misoriented. Were these phenotypes analyzed in Pcdhα-KD tissue? Migrating Pcdhα-KD neurons look bipolar in Figure 1, but there are no Lucida drawings. Likewise, does electroporating the constitutive Rac1 mutant rescue cortical defects in Pcdhα-KD? Again, comparing the same phenotypes across the manipulations would strengthen study's objective that a Pcdhα-WRC-Pyk2 axis regulates neuronal migration and cytoskeletal dynamics.

We have added Lucida drawings of αKD neurons, showing the stunted morphology. As stated above, we have changed the sentence from: “The αKD multipolar neurons in IZ display stunted neurites (Figure 1E)” to: “The αKD multipolar neurons in IZ display stunted processes, as shown by lucida drawings (Figure 1E)”. During migration, cortical neurons reach the lower IZ and become multipolar, starting “multipolar migration” at IZ. The multipolar neurons later reorient the Golgi towards the pia at the IZ/CP and establish a dominant pia-directed leading process, this is known as “multipolar-bipolar transition”, leading to radial migration at CP (Cooper, 2013). Since αKD neurons accumulate in the lower IZ with stunted multipolar morphology, not look bipolar, as shown by Lucida drawings. By contrast, Pyk2OE neurons are stalled in the middle IZ (mIZ) with branchy morphology as discussed above, suggesting defects of multipolar-bipolar transition. The Golgi orientation experiment is designed to address the “multipolar-bipolar transition” defect, so we didn’t analyze the Golgi orientation in αKD neurons.

We have performed the suggested rescue experiments of αKD by electroporating Rac1^Q61L^ and of Pyk2OE by electroporating WAVE2. As shown in Author response image 6, they cannot rescue. This is consistent with our model that they do not have a direct relationship.

**Author response image 6. respfig6:** Rescue experiments of αKD+Rac1^Q61L^ and Pyk2OE+WAVE2.

6) In subsection “Dissection of Pyk2 domain in cortical neuron migration”, the authors describe the Pyk2 structure-function analyses in terms 'recapitulating the migration defects of αKD'. 'Recapitulate' is misused here. Do they mean phenocopy? If that is the case, there is not sufficient evidence for this. As stated in point 5, the phenotypes were not evaluated in the same way, and Pyk2OE leads to multipolar phenotypes that is not shown for αKD. While the Pyk2 structure function analyses do reveal relevant domains, I fail to see how they inform on αKD regulation of Pyk2 activity. Moreover, these manipulations were limited to Pyk2, and were not coupled with αKD manipulations [I initially expected co-transfection experiments but found no description of this approach].

As discussed extensively above, although both αKD and Pyk2OE have cortical neuron migration defect, they are not exactly the same. αKD neurons stalled at lower IZ and have stunted multipolar morphology. Pyk2OE neurons stalled at middle IZ and have branchy morphology. Thus, we do not mean “recapitulate” as “phenocopy”. It’s known that Pyk2 is negatively regulated by Pcdhα (Chen et al., 2009). Thus, αKD results in increased endogenous Pyk2 activity. The suggested overexpression of various Pyk2 dissection experiments would not provide additional new information.

7) The migration defects are vaguely described in the Results section and are not sufficiently discussed in the context of previous studies. Many phenotypes are described, but the biological significances of these effects and whether the components produce the same effects are unclear.For example, in subsection “A role of Pyk2 in cortical neuron migration”, the authors note that Pyk2OE leads to a multipolar phenotype. They interpret the results:"Thus, Pyk2OE blocks multipolar migration by disrupting proper localization of the Golgi apparatus", but this statement is not fully supported by the data, nor are relevant citations given. Jossin et al., showed that Golgi orientation is important for orientating the direction of migration, but does not affect the speed. Here, Pyk2OE also affects migration speed.

As stated above, during the multipolar-bipolar transition, the multipolar cell reorients the Golgi apparatus towards the pia, establishes a dominant pia-directed leading process. This is essential for radial migration. We agree that the statement “Pyk2OE blocks multipolar migration by disrupting proper localization of the Golgi apparatus” is not fully supported by the data. However, Pyk2OE leads to dysregulated Rac1 activity (Suo et al., 2012), resulting in aberrant Arp2/3 and actin assembly. The Golgi apparatus thus cannot reorient. In the end, this results in blocked multipolar-to-bipolar transition. Although Golgi misorientation per se does not affect the speed, the overall radial migration speed is still reduced by Pyk2OE because the cycling of actin fibers at the leading and trailing edges may also be disrupted (in addition to misoriented Golgi).

Regarding this point: " Finally, early born Pyk2OE neurons are also stalled at IZ, suggesting that Pyk2 also plays a role in somal translocation (Figure 3—figure supplement 1C and 1D)."The relevant example is presented in Figure 3H, in the live imaging. But no quantifications are presented.

The quantification of early born Pyk2OE neurons stalling at IZ is shown in Figure 3—figure supplement 1D in the original manuscript. In addition, the quantifications of live imaging of Figure 3H is shown in Figure 3I in the original manuscript.

Xie et al., 2013 showed that during the radial migration phase, cortical neurons undergo a multipolar-bipolar transition in their morphology for glia-guided locomotion, which is dependent on WAVE2, Abi2 (Xie et al., 2013). Is this relevant to the Pcdhα-WAVE-Pyk2 pathway? This could be expanded in the Discussion section.

We appreciate Reviewer #3 for this insightful suggestion. We have expanded the following text in the Discussion section: “The WIRS motif of members of the Pcdhα family binds to a composite surface formed by Abi2 and Sra1 of WAVE (Chen et al., 2014). In addition, the Pcdhα proteins may also recruit WAVE through the direct binding of Abi2 C-terminal SH3 domain to the four PXXP motifs, which are specific to the constant domain of the Pcdhα but not Pcdhγ family (Wu and Maniatis, 1999). Consistently, WAVE2 and Abi2 are required for growth cone activity during cortical neuron migration (Xie et al., 2013).”

8) The sequence of results are disconnected and the rationales are not clear.For example, the transitions between Pcdhα to Pyk2 and back to Pcdhα are disconnected, and the two seem like separate studies.Subsection “A role of Pyk2 in cortical neuron migration”“We previously found that Pcdhα regulates dendritic and spine morphogenesis through inhibiting Pyk2 kinase activity (Suo et al., 2012). To this end, we investigated whether Pyk2 was involved in Pcdhs-regulated cortical neuron migration.”The authors could better articulate their goal to test whether there is increased Pyk2 in αKD tissue, and provide experiments combining KD and Pyk2 manipulations.

Thanks for the good suggestion. We have changed the relevant text to: “Pcdhα physically interacts with and negatively regulates the Pyk2 kinase (Chen et al., 2009). In addition, we previously found that Pcdhα regulates dendritic and spine morphogenesis through inhibiting Pyk2 activity (Suo et al., 2012). To this end, we investigated whether knockdown of Pyk2 could rescue cortical neuron migration defects of αKD.”

Another example: It's not clear why the authors chose to further analyze the functional interactions between Pcdhα and Abi/Wave in lamellipodia using cultures of naïve, non-polarized neurons. Why not revisit this more mature lamellipodia structures relevant to migration, such as leading processes. The data on 'primary neurites' in Figure 5—figure supplement 1 are more convincing than those in Figure 5.

Thanks for this suggestion. We agree that “The data on 'primary neurites' in Figure 5—figure supplement 1 are more convincing than those in Figure 5.” The WAVE2 and Abi2 data in Figure 5—figure supplement 1 are really good. However, the paper is too long. We would be happy to change Figure 5—figure supplement 1 to a main figure if your esteemed journal requires us to do so.

9) The method used for quantifying lamellipodia should be described. Which fluorescent structure was used? (GFP? F-Actin?). The GFP appears to be cytosolic and may not fully resolve the membrane structures.

We used both GFP (labeling transfection) and phalloidin (F-actin) for quantifying lamellipodial structure. We have added the following sentences to Materials and methods section to address these comments: “For quantification, we selected neurons with typical stage1 or stage2 morphology based on GFP and phalloidin signals. For stage1 neurons, we selected the lamellipodia region by the wand tool of the ImageJ software (NIH) and measured the area size. For stage2 neurons, the neurite tips with F-actin-enriched protrusions two folds larger than its width were defined as “neurite with lamellipodia”.

10) Samples sizes are not properly described. Ns are given, but it's not clear what they comprise. The authors should report the numbers of cells, sections analyzed, animals per litter, numbers of litter/replicates.

We have added the detailed information as described above.

11) Statistical tests are limited to t-tests. Bu in many instances, multiple groups are analyzed and thus ANOVA and post-hoc multiple comparisons would be more suitable.

We have now used ANOVA and post-hoc multiple comparisons to analyze multiple groups.

12) I think it is also premature to exclude effects on survival, since quantification of numbers of GFP labeled cells in each IUE manipulation are not given. Cleaved-caspase 3 might only be detected during a small window, and labeled cells are cleared rapidly.

We have quantified the numbers of GFP labeled cells in each IUE manipulation. As shown in Author response image 7, there is no statistical difference between scrambled (SCR) and α knockdowns (αKD-1 and αKD-2). Consistent with this quantification, cleaved-caspase 3 staining did not reveal increased apoptosis. Although cleaved-caspase 3 might only be detected during a small window, and labeled cells are cleared rapidly, this method has been used in many studies to detect apoptotic cells such as Nancy Ip’s recent work on cortical neuron migration (Ye et al., 2014).

**Author response image 7. respfig7:** Quantification of the numbers of GFP labeled cells in each IUE manipulation. One-way ANOVA, ns, not significant.